# Online Learning in Periodic Zero-Sum Games

**Tanner Fiez**[*]
University of Washington
Seattle, Washington
fiezt@uw.edu

**Ryann Sim**[*]
SUTD
Singapore
ryann_sim@mymail.sutd.edu.sg

**Stratis Skoulakis**[*]
SUTD
Singapore
efstratios@sutd.edu.sg

**Georgios Piliouras**[†]
SUTD
Singapore
georgios@sutd.edu.sg

**Lillian Ratliff**[†]
University of Washington
Seattle, Washington
ratliffl@uw.edu

## Abstract

A seminal result in game theory is von Neumann's minmax theorem, which states that zero-sum games admit an essentially unique equilibrium solution. Classical learning results build on this theorem to show that online no-regret dynamics converge to an equilibrium in a time-average sense in zero-sum games. In the past several years, a key research direction has focused on characterizing the day-to-day behavior of such dynamics. General results in this direction show that broad classes of online learning dynamics are cyclic, and formally Poincaré recurrent, in zero-sum games. We analyze the robustness of these online learning behaviors in the case of periodic zero-sum games with a time-invariant equilibrium. This model generalizes the usual repeated game formulation while also being a realistic and natural model of a repeated competition between players that depends on exogenous environmental variations such as time-of-day effects, week-to-week trends, and seasonality. Interestingly, time-average convergence may fail even in the simplest such settings, in spite of the equilibrium being fixed. In contrast, using novel analysis methods, we show that Poincaré recurrence provably generalizes despite the complex, non-autonomous nature of these dynamical systems.

## 1 Introduction

The study of learning dynamics in zero-sum games is arguably as old of a field as game theory itself, dating back to the seminal work of Brown and Robinson [7, 27], which followed shortly after the foundational minmax theorem of von Neumann [33]. The dynamics of online no-regret learning algorithms [10, 29] are of particular interest in zero-sum games as they are designed with an adversarial environment in mind. Moreover, well known results imply that such dynamics converge in a time-average sense to a minmax equilibrium in zero-sum games [10, 15].

Despite the classical nature of the study of online no-regret learning dynamics in zero-sum games, the actual transient behavior of such dynamics was historically not as understood. However, in the past several years this topic has gained attention with a number of works studying such dynamics in zero-sum games (and variants thereof) with a particular focus on continuous-time analysis [24, 25, 20, 6, 32, 23, 22]. The unifying emergent picture is that the dynamics are "approximately cyclic" in a formal sense known as Poincaré recurrence. Moreover, these results have acted as fundamental building blocks for understanding the limiting behavior of their discrete-time variants [2, 12, 21, 11, 3].

---

[*]Joint first authors

[†]Joint last authors

35th Conference on Neural Information Processing Systems (NeurIPS 2021).

Despite the plethora of emerging results regarding online learning dynamics in zero-sum games, an important and well motivated aspect of this problem has only begun to receive attention.

*How do online learning dynamics behave if the zero-sum game evolves over time?*

Clearly, the answer to this question depends critically on how the game is allowed to evolve.

**Problem Setting and Model.** We study periodic zero-sum games with a time-invariant equilibrium, which is a class of games we formally define in Section 2. In a periodic zero-sum game, the payoffs that dictate the game are both $T$-periodic and zero-sum at all times. We consider both periodic zero-sum bilinear games on infinite, unconstrained strategy spaces and periodic zero-sum matrix games (along with network generalizations thereof) on finite strategy spaces. The goal of this work is to evaluate the robustness of the archetypal online learning behaviors in zero-sum games, Poincaré recurrence and time-average equilibrium convergence, to this natural model of game evolution.

**Connections to Repeated Game Models.** The time-evolving game model we study can be seen as a generalization of usual repeated game formulations. A time-invariant game is a trivial version of a periodic game, in which case we recover the repeated static game setting. For a general periodic zero-sum game with period $T$, each stage game now is chosen according to a fixed length $T$ sequence of games, capturing interactions between the players with time-dependent payoffs.

Periodic zero-sum games can also fit into the frameworks of multi-agent contextual games [28] and dynamic games (see, e.g., [5]). In a multi-agent contextual game [28], the environment selects a context from a set before each round of play and this choice defines the game that is played. Periodic zero-sum games can be seen as a multi-agent contextual game where the environment draws contexts from the available set in a $T$-periodic fashion with each context defining a zero-sum game with a common equilibrium. In the class of dynamic games, there is a game state on which the payoffs may depend that evolves with dynamics. Periodic zero-sum games can be interpreted as a dynamic game where the state transitions do not depend on the strategies of the players, the state is $T$-periodic, and the payoffs are completely defined by the state. We remark that the focus of existing work on contextual games and dynamic games is distinct from the questions investigated in this paper.

The periodic zero-sum game model allows us to capture competitive settings where exogenous environmental variations manifest in an effectively periodic/epochal fashion. This naturally occurs in market competitions where time-of-day effects, week-to-week trends, and seasonality can dictate the game between players. To illustrate this point, consider a competition between service providers that wish to maximize their users, while the total market size evolves seasonally over time. This evolution affects the utility functions, even if the fundamentals of the market, and consequently the equilibrium, remain invariant.

**Contributions and Approach.** In this paper, for the classes of periodic zero-sum bilinear games and periodic zero-sum polymatrix games with time-invariant equilibrium, we investigate the day-to-day and time-average behaviors of continuous-time gradient descent-ascent (GDA) and follow-the-regularized-leader (FTRL) learning dynamics, respectively. This study highlights the careful attention that must be given to the dynamical systems in periodic zero-sum games which preclude standard proof techniques for Poincaré recurrence, while also revealing that intuition from existing results on static zero-sum games can be totally invalidated even by simple restricted examples in periodic zero-sum games.

**Contribution 1: Poincaré Recurrence.** A key technical challenge in this work is that the dynamical systems which emerge from learning dynamics in periodic zero-sum games correspond to *non-autonomous* ordinary differential equations, whereas learning dynamics in static zero-sum games correspond to *autonomous* ordinary differential equations. Consequently, the usual proof methods from static zero-sum games for showing Poincaré recurrence are insufficient on their own in periodic zero-sum games. We overcome this challenge by delicately piecing together properties of periodic systems to construct a discrete-time autonomous system that we are able to show is Poincaré recurrent. This approach allows to prove both the GDA and FTRL learning dynamics are Poincaré recurrent in the respective classes of periodic zero-sum games (Theorems 1 & 2). Finally, we show both periodicity and a time-invariant equilibrium are necessary for such results in evolving games (Proposition 1).

**Contribution 2: Time-Average Strategy Equilibration Fails.** Given that Poincaré recurrence provably generalizes from static zero-sum games to periodic zero-sum games, it may be expected that the time-average strategies in periodic zero-sum games converge to the time-invariant equilibrium

as in static zero-sum games. Surprisingly, we show that counterexamples can be constructed to this intuition even in the simplest of periodic zero-sum games. In particular, we prove the negative result that the time-average GDA and FTRL strategies do not necessarily converge to the time-invariant equilibrium in the respective classes of zero-sum games (Propositions 2 & 3).

**Contribution 3: Time-Average Equilibrium Utility Convergence.** Despite the negative result on the time-average strategy convergence, in the special case of periodic zero-sum bimatrix games we are able to show a complimentary positive result on the time-average utility convergence. Specifically, we show that the time-average utilities of the FTRL learning dynamics converge to the average of the equilibrium utility values of all the zero-sum games included in a single period of our time-evolving games. (Theorem 3).

**Organization.** In Section 2, we formalize the classes of games that we study. We present characteristics of dynamical systems as they pertain to this work in Section 3. Section 4 and 5 contain our results analyzing GDA and FTRL learning dynamics in continuous and finite strategy periodic zero-sum bilinear and polymatrix games, respectively. We present numerical experiments in Section 6 and finish with a discussion in Section 7. Proofs of are theoretical results are deferred to the appendix.

## 2  Game-Theoretic Preliminaries

### 2.1  Continuous Strategy Periodic Zero-Sum Games

For continuous strategy periodic zero-sum games, we study periodic zero-sum bilinear games. We begin by formalizing zero-sum bilinear games and then define the periodic variant.

**Zero-Sum Bilinear Games.** Given a matrix $A \in \mathbb{R}^{n_1 \times n_2}$, a zero-sum bilinear game on continuous strategy spaces can be defined by the max-min problem $\max_{x_1 \in \mathbb{R}^{n_1}} \min_{x_2 \in \mathbb{R}^{n_2}} x_1^\top A x_2$. Formally, the game is defined by the pair of payoff matrices $\{A, -A^\top\}$ and the action space of agents 1 and 2 are given by $\mathbb{R}^{n_1}$ and $\mathbb{R}^{n_2}$, respectively. Player 1 seeks to maximize the utility function $u_1(x_1, x_2) = x_1^\top A x_2$ while player 2 optimizes the utility $u_2(x_1, x_2) = -x_2^\top A^\top x_1$. The game is zero-sum since for any $x_1 \in \mathbb{R}^{n_1}$ and $x_2 \in \mathbb{R}^{n_2}$, the sum of utility over each player is zero. For zero-sum bilinear games, a *Nash equilibrium* corresponds to a joint strategy $(x_1^*, x_2^*)$ such that for each player $i$ and $j \neq i$, $u_i(x_i^*, x_j^*) \geq u_i(x_i, x_j^*)$, $\forall x_i \in \mathbb{R}^{n_i}$. Note that $(x_1^*, x_2^*) = (\mathbf{0}, \mathbf{0})$ is always a Nash equilibrium of a zero-sum bilinear game.

**Periodic Zero-Sum Bilinear Games.** We study the continuous-time GDA learning dynamics in a class of games we refer to as periodic zero-sum bilinear games. The key distinction from a typical static zero-sum bilinear game is that the payoff matrix is no longer fixed in this class of games. Instead, the payoff matrix may change at each time instant as long as game remains zero-sum and the continuous-time sequence of payoffs is periodic. The next definition formalizes this class of games.

**Definition 1** (Periodic Zero-Sum Bilinear Game). *A periodic zero-sum bilinear game is an infinite sequence of zero-sum bilinear games $\{A(t), -A(t)^\top\}_{t=0}^\infty$ in which the player set and strategy spaces are fixed and the payoff matrix is such that $A(t) = A(t+T)$ for a finite period $T$ and all $t \geq 0$. Note that in such a game, $(0,0)$ is always a time-invariant Nash equilibrium. Furthermore, we assume that the dependence of the payoff entries on time is smooth everywhere except for a finite set of points.*

### 2.2  Finite Strategy Periodic Zero-Sum Games

For finite strategy periodic zero-sum games, we analyze periodic zero-sum polymatrix games. In what follows we define a zero-sum polymatrix game, which is a network generalization of a bimatrix game, and then detail the periodic variant considered in this paper.

**Zero-Sum Polymatrix Games.** An $N$-player polymatrix game is defined by an undirected graph $G = (V, E)$ where $V$ is the player set and $E$ is the edge set where a bimatrix game is played between the endpoints of each edge [8]. Each player $i \in V$ has a set of actions $\mathcal{A}_i = \{1, \ldots, n_i\}$ that can be selected at random from a distribution $x_i$ called a mixed strategy. The mixed strategy set of player $i \in V$ is the simplex in $\mathbb{R}^{n_i}$ denoted by $\mathcal{X}_i = \Delta^{n_i - 1} = \{x_i \in \mathbb{R}^{n_i}_{\geq 0} : \sum_{\alpha \in \mathcal{A}_i} x_{i\alpha} = 1\}$ where $x_{i\alpha}$ denotes the probability of action $\alpha \in \mathcal{A}_i$. The joint strategy space is denoted by by $\mathcal{X} = \Pi_{i \in V} \mathcal{X}_i$.

The bimatrix game on edge $(i, j)$ is described using a pair of matrices $A^{ij} \in \mathbb{R}^{n_i \times n_j}$ and $A^{ji} \in \mathbb{R}^{n_j \times n_i}$. The utility or payoff of agent $i \in V$ under the strategy profile $x \in \mathcal{X}$ is given by $u_i(x) =$

$\sum_{j:(i,j)\in E} x_i^\top A^{ij} x_j$ and corresponds to the sum of payoffs from the bimatrix games the player participates in. We further denote by $u_{i\alpha}(x) = \sum_{j:(i,j)\in E}(A^{ij}x_j)_\alpha$ the utility of player $i \in V$ under the strategy profile $x = (\alpha, x_{-i}) \in \mathcal{X}$ for $\alpha \in \mathcal{A}_i$. The game is called zero-sum if $\sum_{i \in V} u_i(x) = 0$ for all $x \in \mathcal{X}$. Each bimatrix edge game is not necessarily zero-sum in a zero-sum polymatrix game.

A *Nash equilibrium* in a polymatrix game is a mixed strategy profile $x^* \in \mathcal{X}$ such that for each player $i \in V$, $u_i(x_i^*, x_{-i}^*) \geq u_i(x_i, x_{-i}^*)$, $\forall x_i \in \mathcal{X}_i$. A Nash equilibrium is said to be an interior if $\mathrm{supp}(x_i^*) = \mathcal{A}_i \; \forall i \in V$ where $\mathrm{supp}(x_i^*) = \{\alpha \in \mathcal{A}_i : x_{i\alpha} > 0\}$ is the support of $x_i^* \in \mathcal{X}_i$.

**Periodic Zero-Sum Polymatrix Games.** We analyze the continuous-time FTRL learning dynamics in a class of games we call periodic zero-sum polymatrix games. This class of games is such that the payoffs defined by the edge games evolve periodically. We consider that this periodic evolution is such that there is a common interior Nash equilibrium that arises in each zero-sum polymatrix game that arrives. The following definition formalizes the games we study on finite strategy spaces.

**Definition 2** (Periodic Zero-Sum Polymatrix Game). *A periodic zero-sum polymatrix game is an infinite sequence of zero-sum polymatrix games $\{G(t) = (V(t), E(t))\}_{t=0}^\infty$ in which the set of players, strategy spaces, and edges are fixed and each bimatrix game on an edge $(i,j)$ is such that $A^{ij}(t) = A^{ij}(t+T)$ and $A^{ji}(t) = A^{ji}(t+T)$ for some finite period $T$ and all $t \geq 0$. We assume there is a common interior Nash equilibrium $x^* \in \mathcal{X}$ of the polymatrix game $G(t)$ for all $t \geq 0$. Furthermore, we assume that the dependence of the payoff entries on time is smooth everywhere except for a finite set of points.*

## 3 Preliminaries on Dynamical Systems

We now cover concepts from dynamical systems theory that will help us analyze learning dynamics in periodic zero-sum games and prove Poincaré recurrence. Careful attention must be given to these preliminaries in this work since the dynamical systems we study are non-autonomous whereas typical recurrence analysis in the study of learning in games deals with autonomous dynamical systems.

### 3.1 Background on Dynamical Systems

We begin this section by providing dynamical systems background that is necessary both for defining Poincaré recurrence and sketching typical proof methods along with our approach.

**Flows.** Consider an ordinary differential equation $\dot{x} = f(t,x)$ on a topological space $X$. We can define the *flow* $\phi : \mathbb{R} \times X \to X$ of a dynamical system $\dot{x}$, for which the following holds: (i) $\phi(t,\cdot) : X \to X$, often denoted $\phi^t : X \to X$, is a homeomorphism for each $t \in \mathbb{R}$, (ii) $\phi(t+s, x) = \phi(t, \phi(s,x))$ for all $t, s \in \mathbb{R}$ and all $x \in X$, (iii) for each $x \in X$, $\frac{d}{dt}|_{t=0}\phi(t,x) = f(t,x)$, and (iv) $\phi(t, x_0) = x(t)$ is the solution.

**Existence and Uniqueness.** We utilize Carathéodory's existence theorem to guarantee the existence of a flow for $\dot{x}$, even for some discontinuous functions $f$.

**Theorem** (Carathéodory's existence theorem [13, 16]). *Consider a differential equation $\dot{x} = f(t,x)$ on a rectangular domain $R = \{(t,y)| \; |t - t_0| \leq a, |x - x_0| \leq b\}$. If $f$ satisfies the following conditions:*

1. *$f(t,x)$ is continuous in $y$ for each fixed $t$,*

2. *$f(t,x)$ is measurable in $t$ for each fixed $y$,*

3. *there is a Lebesgue-integrable function $m : [t_0 - a, t_0 + a] \to [0,\infty)$ such that $|f(t,x)| \leq m(t)$ for all $(t,x) \in R$,*

*then the differential equation has a solution. Moreover, if $f$ is also Lipschitz continuous, meaning $|f(t,x_1) - f(t,x_2)| \leq k(t)|x_1 - x_2|$ with some Lebesgue-integrable function $k : [t_0 - a, t_0 + a] \to [0,\infty)$, then there exists a unique solution of the differential equation.*

In the settings we study, the above three conditions hold: Condition 1 holds because for every fixed $t$, the dynamics we study (specifically GDA and FTRL) are continuous functions of their state space. Condition 2 holds because the systems we study are finite and continuous almost-everywhere, and

so by Lusin's theorem [18] are measurable for each fixed $y$. Finally, Condition 3 is always satisfied because the games we study always admit bounded orbits. Hence, it follows that a unique flow exists for all the dynamical systems studied in this paper.

**Conservation of Volume.** The flow $\phi$ of an ordinary differential equations is called *volume preserving* if the volume of the image of any set $U \subseteq \mathbb{R}^d$ under $\phi^t$ is preserved, meaning that $\text{vol}(\phi^t(U)) = \text{vol}(U)$. *Liouville's theorem* states that a flow is volume preserving if the divergence of $f$ at any point $x \in \mathbb{R}^d$ equals zero: that is, $\text{div} f(t, x) = \text{tr}(Df(t, x)) = \sum_{i=1}^{d} \frac{\partial f(t, x)}{\partial x_i} = 0$.

We now transition to give general Poincaré recurrence statements along with discussion of how the results are usually applied in game theory before we outline our proof methods.

### 3.2 Poincaré Recurrence in Autonomous Dynamical Systems

A number of works in the past several years show that online no-regret learning dynamics are Poincaré recurrent in repeated static zero-sum games (see, e.g., [24, 20, 6]). The proof methods for deriving such results crucially rely on the static nature of the game for the reason that the learning dynamics amount to an autonomous dynamical system. Informally, the standard Poincaré recurrence theorem states that if an autonomous dynamical system preserves volume and every orbit remains bounded, almost all trajectories return arbitrarily close to their initial position, and do so infinitely often [26]; this property of a dynamical system is known as *Poincaré recurrence*. Thus, proving the Poincaré recurrence of dynamics in repeated static zero-sum games is tantamount to verifying the volume preservation and bounded orbit properties. We now formalize the Poincaré recurrence theorem.

Given a flow $\phi^t$ on a topological space $X$, a point $x \in X$ is *nonwandering* for $\phi^t$ if for each open neighborhood $U$ containing $x$, there exists $T > 1$ such that $U \cap \phi^T(U) \neq \emptyset$. The set of all nonwandering points for $\phi^t$, called the *nonwandering set*, is denoted $\Omega(\phi^t)$.

**Theorem** (Poincaré Recurrence for Continuous-Time Systems [26])**.** *If a flow preserves volume and has only bounded orbits, then for each open set almost all orbits intersecting the set intersect it infinitely often: if $\phi^t$ is a volume preserving flow on a bounded set $Z \subset \mathbb{R}^d$, then $\Omega(\phi^t) = Z$.*

In order to describe our proof methods in the following subsection for showing Poincaré recurrence in periodic zero-sum games, it will be useful for us to state an alternative formulation of the Poincaré recurrence theorem that is applicable to autonomous discrete-time systems.

**Theorem** (Poincaré Recurrence for Discrete-Time Maps [4])**.** *Let $(X, \Sigma, \mu)$ be a finite measure space and let $\phi\colon X \to X$ be a measure-preserving map. For any $E \in \Sigma$, the set of those points $x$ of $E$ for which there exists $N \in \mathbb{N}$ such that $\phi^n(x) \notin E$ for all $n > N$ has zero measure. In other words, almost every point of $E$ returns to $E$. In fact,* almost every point returns infinitely often. *That is, $\mu(\{x \in E : \exists N \text{ s.t. } \phi^n(x) \notin E \text{ for all } n > N\}) = 0$.*

### 3.3 Poincaré Recurrence in Periodic Dynamical Systems

The proof methods described in the last section for showing the Poincaré recurrence of dynamics in static zero-sum games cannot directly be applied to time-evolving zero-sum games as a result of the non-autonomous nature of the systems. In fact, we can construct time-evolving zero-sum games without both periodic payoffs and a time-invariant equilibrium, where online learning dynamics are not Poincaré recurrent.[3] Despite this hurdle, we show that in the natural subclass of time-evolving games covered by periodic zero-sum games (periodic payoffs and a time-invariant equilibrium), we can develop proof methods to show the Poincaré recurrence of online learning dynamics. Given the previous claim regarding the possible non-recurrent nature of learning dynamics when there is not both periodic payoffs and a time-invariant equilibrium, this is perhaps the most general class of time-evolving zero-sum games with obtainable positive results in this direction.

We now give an overview of our approach, beginning by recalling properties of periodic systems.

**Periodic Systems and Poincaré Maps.** A system $\dot{x} = f(t, x)$ is $T$-periodic if $f(t + T, x) = f(t, x)$ for all $(x, t)$. Let $\phi^t : \mathbb{R}^n \to \mathbb{R}^n$ denote the mapping taking $x \in \mathbb{R}^n$ to the value at time $t$. For a $T$-periodic system, $\phi^{T+s} = \phi^s \circ \phi^T$ so that $\phi^{kT} = (\phi^T)^k$ for any integer $k$. The mapping $\phi^T : \mathbb{R}^n \to \mathbb{R}^n$ is called the *Poincaré map* or the *mapping at a period*.

---

[3]We formalize this statement in Proposition 1 of the following section.

If the differential equation is well-defined for all $x$ and has a solution for all $t \in [0, T]$, then for each initial condition (where we have suppressed the dependence on $x_0$), the Poincaré map $\phi^T$ defines a discrete-time autonomous dynamical system $x^+ = \phi^T(x)$. The learning dynamics we study in periodic zero-sum games form $T$-periodic dynamical systems. Thus, the discrete-time autonomous dynamical system $x^+ = \phi^T(x)$ formed by the Poincaré map is key to the analysis methods we pursue. In particular, our approach is to show that this system is Poincaré recurrent, which we then use to conclude that the original continuous-time non-autonomous system is Poincaré recurrent.

Given the previously presented Poincaré recurrence theorem for discrete-time maps, proving the Poincaré recurrence of the system $x^+ = \phi^T(x)$ requires verifying the volume preservation and bounded orbit properties, which then implies the measure preserving property. The following result states that if the divergence of a $T$-periodic vector field $f(x, t)$ is divergence free so that the flow $\phi^t$ is volume preserving, then the Poincaré map $\phi^T$ and the resulting discrete-time dynamical system $x^+ = \phi^T(x)$ is also volume preserving.

**Theorem** (Volume preservation for $T$-Periodic Systems [1, 3.16.B, Thm 2]). *If the $T$-periodic system $\dot{x} = f(t, x)$ is divergence-free, then $\phi^T$ preserves volume.*

Similarly, if orbits of $\dot{x} = f(t, x)$ are bounded, then clearly the orbits of $x^+ = \phi^T(x)$ are bounded. Hence, to show the system $x^+ = \phi^T(x)$ is Poincaré recurrent, we prove $\dot{x} = f(t, x)$ has a divergence-free vector-field (equivalently, that the flow is volume preserving) and only bounded orbits. This will then be sufficient to conclude $\dot{x} = f(t, x)$ is Poincaré recurrent since the discrete-time system forms a subsequence of the continuous-time system.

## 4 Gradient Descent-Ascent in Periodic Zero-Sum Bilinear Games

This section focuses on the continuous-time GDA learning dynamics in periodic zero-sum bilinear games. The dynamics are such that each player seeks to maximize their utility by following the gradient with respect to their choice variable and are given by

$$\dot{x}_1 = A(t)x_2(t)$$
$$\dot{x}_2 = -A^\top(t)x_1(t).$$

### 4.1 Poincaré Recurrence

The focus of this section is on characterizing the transient behavior of the continuous-time GDA learning dynamics in periodic zero-sum games. Specifically, we show the following result.

**Theorem 1.** *The continuous-time* GDA *learning dynamics are Poincaré recurrent in any periodic zero-sum bilinear game as given in Definition 1.*

Theorem 1 establishes that the recurrent nature of continuous-time GDA dynamics in static zero-sum bilinear games is robust to the dynamic evolution of the payoffs in periodic zero-sum bilinear games.

Prior to outlining the proof steps for Theorem 1, we elaborate on the claim from the previous section that without the periodicity property and a time-invariant equilibrium, such a result is unobtainable. In particular, we show the Poincaré recurrence of the GDA dynamics is not guaranteed without both properties by constructing counterexamples when only one of the properties holds.

**Proposition 1.** *There exists time-evolving zero-sum games such that there is a time-invariant equilibrium or the payoffs are periodic (but not both simultaneously) in which the* GDA *dynamics are not Poincaré recurrent.*

This proposition highlights the strength of our results regarding GDA, given that the assumptions needed to obtain them are more or less tight.

We now outline the key intermediate results we prove to obtain Theorem 1, following the techniques described in Section 3.3. For the GDA dynamics, we utilize the observation that the corresponding vector fields are divergence free to show that the learning dynamics are volume-preserving. We now state this result formally.

**Lemma 1.** *The* GDA *learning dynamics are volume preserving in any periodic zero-sum bilinear game as given in Definition 1.*

We then proceed by showing that the GDA orbits are bounded by deriving a time-invariant function. This step relies on the fact that we have a time-invariant equilibrium.

**Lemma 2.** *The function* $\Phi(t) = \frac{1}{2}\left(x_1^\top(t)x_1(t) + x_2^\top(t)x_2(t)\right)$ *is time-invariant. Hence, the* GDA *orbits are bounded in any periodic zero-sum bilinear game as given in Definition 1.*

Given the volume preservation and bounded orbit characteristics of the continuous-time GDA learning dynamics in periodic zero-sum games, the proof of recurrence follows by applying the arguments described in Section 3.3.

## 4.2 Time-Average Convergence

The Poincaré recurrence continuous-time GDA learning dynamics in periodic zero-sum bilinear games indicates that the system has regularities which couple the evolving players and evolving game despite the failure to converge to a fixed point. A natural follow-up question to the cyclic transient behavior of the dynamics is whether the long-run converges to a game-theoretically meaningful outcome.

We show that in periodic zero-sum bilinear games, the time-average of GDA learning dynamics may not converge to the time-invariant Nash equilibrium. To prove this, we consider a periodic zero-sum bilinear game with the action space of each player on $\mathbb{R}$ so that the evolving payoff simply rescales the vector field. We construct the payoff sequence so that the dynamics return back to the initial condition after a period of the game, while the time-average of the dynamics are not equal to the time-invariant equilibrium $(x_1^*, x_2^*) = (\mathbf{0}, \mathbf{0})$. Given the simplicity of this example, it effectively rules out hope to provide a meaningful time-average convergence guarantee in this class of games.

**Proposition 2.** *There exists periodic zero-sum bilinear games satisfying Definition 1 where the time-average strategies of the* GDA *dynamics fail to converge to the time-invariant equilibrium* $(\mathbf{0}, \mathbf{0})$.

# 5 Follow-the-Regularized-Leader in Periodic Zero-Sum Polymatrix Games

We now analyze continuous-time FTRL learning dynamics in periodic zero-sum polymatrix games. Players that follow FTRL learning dynamics in this class of games select a mixed strategy at each time that maximizes the difference between the cumulative payoff evaluated over the history of games and a regularization penalty. This adaptive strategy balances exploitation based on the past with exploration.

Formally, the continuous-time FTRL learning dynamics for any player $i \in V$ in a periodic zero-sum polymatrix game with an initial payoff vector $y_i(0) \in \mathbb{R}^{n_i}$ are given by

$$y_i(t) = y_i(0) + \int_0^t \sum_{j:(i,j)\in E} A^{ij}(\tau)x_j(\tau)d\tau \tag{1}$$
$$x_i(t) = \mathrm{argmax}_{x_i \in \mathcal{X}_i}\{\langle x_i, y_i(t)\rangle - h_i(x_i)\}$$

where $h_i : \mathcal{X}_i \to \mathbb{R}$ is a penalty term which encourages exploration away from the strategy which maximizes the cumulative payoffs in hindsight. We assume that the regularization function $h_i(\cdot)$ for each player $i \in V$ is continuous, strictly convex on $\mathcal{X}_i$, and smooth on the relative interior of every face of $\mathcal{X}_i$. These assumptions ensure the update $x_i(t)$ is well-defined since a unique solution exists.

Common FTRL learning dynamics include the multiplicative weights update and the projected gradient dynamics. The multiplicative weights dynamics for a player $i \in V$ arise from the regularization function $h_i(x_i) = \sum_{\alpha \in \mathcal{A}_i} x_{i\alpha} \log x_{i\alpha}$ and correspond to the replicator dynamics. The projected gradient dynamics for a player $i \in V$ derive from the Euclidean regularization $h_i(x_i) = \frac{1}{2}\|x_i\|_2^2$.

To simplify notation, the FTRL dynamics can equivalently be formulated as the following update

$$y_i(t) = y(0) + \int_0^t v_i(x(\tau), \tau)d\tau \tag{2}$$
$$x_i(t) = Q_i(y_i(t)).$$

Observe that we denote by $v_i(x, \tau) = (u_{i\alpha}(x, \tau))_{\alpha \in \mathcal{A}_i}$ the vector of each pure strategy $\alpha \in \mathcal{A}_i$ utility for agent $i \in V$ under the joint profile $x = (\alpha, x_{-i}) \in \mathcal{X}$ at time $\tau \geq 0$. Moreover, $Q_i : \mathbb{R}^{n_i} \to \mathcal{X}_i$ is known as the choice map and defined as

$$Q_i(y_i(t)) = \mathrm{argmax}_{x_i \in \mathcal{X}_i}\{\langle y_i(t), x_i\rangle - h_i(x_i)\}.$$

In this notation, the utility of the player $i \in V$ under the joint strategy $x = (x_i, x_{-i}) \in \mathcal{X}$ at time $t \geq 0$ is given by $u_i(x, \tau) = \langle v_i(x, \tau), x_i \rangle$. Observe that in our notation of utility we are now including the time index to make the dependence on the evolving game and payoffs explicit.

For any player $i \in V$ we denote by $h_i^* : \mathbb{R}^{n_i} \to \mathbb{R}$ the convex conjugate of the regularization function $h_i : \mathcal{X}_i \to \mathbb{R}$ which is given by the quantity $h_i^*(y_i(t)) = \max_{x_i \in \mathcal{X}_i}\{\langle x_i, y_i(t)\rangle - h_i(x_i)\}$.

## 5.1 Poincaré Recurrence

We now focus on characterizing the transient behavior of the continuous-time FTRL learning dynamics in periodic zero-sum polymatrix games. It is known that the continuous-time FTRL learning dynamics are Poincaré recurrent in static zero-sum polymatrix games [20]. The following result demonstrates that this characteristic holds even in games that are evolving in a periodic fashion with a time-invariant equilibrium, providing a broad generalization.

**Theorem 2.** *The* FTRL *learning dynamics are Poincaré recurrent in any periodic zero-sum polymatrix game as given in Definition 2.*

For the remainder of this subsection, we describe our proof methods. The general approach is that we prove the Poincaré recurrence of a transformed system using the techniques described in Section 3.3. This conclusion then allows us to infer the equivalent property for the original FTRL system.

The utility differences for each player $i \in V$ and pure strategy $\alpha_i \in \mathcal{A}_i \setminus \beta_i$ evolve following the differential equation
$$\dot{z}_{i\alpha_i} = v_{i\alpha_i}(x(t), t) - v_{i\beta_i}(x(t), t).$$
Toward proving that this system is Poincaré recurrent, we show that the vector field $\dot{z}$ is divergence free and hence volume preserving.

**Lemma 3.** *The dynamics defined by the system $\dot{z}$ are volume preserving in any periodic zero-sum polymatrix game as given in Definition 2.*

We then construct a time-invariant function along the evolution of the system that is sufficient to guarantee that the orbits generated by the $\dot{z}$ dynamics are bounded. Recall that $x^*$ denotes the common, time-invariant interior Nash equilibrium of the periodic zero-sum polymatrix game.

**Lemma 4.** *The function $\Phi(x^*, y(t)) = \sum_{i \in V}\left(h_i^*(y_i(t)) - \langle x_i^*, y_i(t)\rangle + h_i(x_i^*)\right)$ is time-invariant. Hence, the orbits generated by the $\dot{z}$ dynamics are bounded in any periodic zero-sum polymatrix game as given in Definition 2.*

From this point, we follow the arguments from Section 3.3 to conclude the $\dot{z}$ dynamics are Poincaré recurrent. Finally, we show that Poincaré recurrence of the $\dot{z}$ system is sufficient to guarantee the Poincaré recurrence of the FTRL learning dynamics.

The proofs of the results in this section can be found in Appendix D.

## 5.2 Time-Average Convergence

A number of well-known properties of zero-sum bimatrix games fail to generalize to zero-sum polymatrix games. Indeed, fundamental characteristics of zero-sum bimatrix games include that each agent has a unique utility value in any Nash equilibrium and that equilibrium strategies are exchangeable. However, Cai and Daskalakis [8] show that neither of these properties are guaranteed in zero-sum polymatrix games. Consequently, in general, time-average convergence to the set of equilibrium values in the utility and strategy spaces does not equate to the stronger notion of pointwise convergence.

For the reasons just outlined, we pursue a different notion of time-average convergence in periodic zero-sum polymatrix games. That is, we consider the subclass of periodic zero-sum bimatrix games (2-player periodic zero-sum polymatrix games) and show that the time-average utility of each agent converges to the time-average of the game values (that is, the unique utility the player obtains at any Nash equilibrium) over a period of the periodic game.

**Theorem 3.** *In periodic zero-sum bimatrix games satisfying Definition 2, if each player follows* FTRL *dynamics, then the time-average utility of each player converges to the time-average over a period of the game equilibrium utility values.*

Theorem 3 paints a positive view of the time-average behavior of FTRL learning dynamics in periodic zero-sum games. However, the following result demonstrates that much like in the case of GDA in periodic zero-sum bilinear games, the time-average strategies are not guaranteed to converge to the time-invariant Nash equilibrium.

**Proposition 3.** *There exist periodic zero-sum bimatrix games satisfying Definition 1 in which the time-average strategies of* FTRL *dynamics fail to converge to the time-invariant Nash equilibrium.*

We prove the negative result in this proposition by constructing a simple counterexample that corresponds to a time-varying rescaling of Matching Pennies.

The proofs of the results in this section can be found in Appendix E.

## 6 Experiments

In this section, we present several experimental simulations that illustrate our theoretical results. To begin, for continuous-time GDA dynamics we show that Poincaré recurrence holds in a periodic zero-sum bilinear game. We consider the ubiquitous Matching Pennies game with payoff matrix $\left(\begin{smallmatrix} 1 & -1 \\ -1 & 1 \end{smallmatrix}\right)$. We then use the following periodic rescaling with period $2\pi$:

$$\alpha(t) = \begin{cases} \sin(t) & 0 \leq t \leq \frac{3\pi}{2} \\ \left(\frac{2}{\pi}\right)(t \bmod(2\pi) - 2\pi) & \frac{3\pi}{2} \leq t \leq 2\pi \end{cases}$$

When players follow the GDA learning dynamics, we see from Figure 1 that the trajectories when plotted alongside the value of the periodic rescaling are bounded. A similar experimental result holds in the case of FTRL dynamics. In the supplementary material, we simulate replicator dynamics with the same periodic rescaling as in Figure 1. The trajectories in the dual/payoff space also remain bounded due to the invariance of the Kullback-Leibler divergences (KL-divergence).

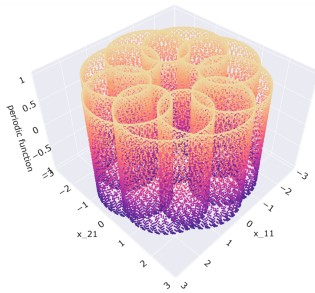

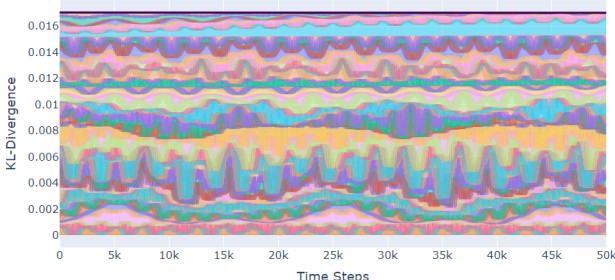

Figure 1: Bounded trajectories for a periodically rescaled Matching Pennies game updated using GDA.

Figure 2: Weighted sum of KL-divergences for a 64-player periodically rescaled Matching Pennies game. Note that despite the complicated trajectories of each player, the weighted sum of their divergences remains constant.

Lemmas 2 and 4 describe functions $\Phi$ which remain time-invariant. In the case of replicator dynamics, $\Phi(t)$ is the sum of Kullback-Leibler divergences measured between the strategy of each player and their mixed Nash strategy $[1/2, 1/2]$. We simulated a 64-player polymatrix extension to the Matching Pennies game, where each agent plays against the opponent immediately adjacent to them, forming a 'toroid'-like chain of games. Furthermore, we randomly rescale each game with a different periodic function. Figure 2 depicts the claim presented in the lemmas: although each agent's specific divergence term $\mathrm{KL}(x_i^* \| x_i(t))$ fluctuates, the sum $\sum_{i \in V} \mathrm{KL}(x_i^* \| x_i(t))$ remains constant.

To generate Figure 3, we show the data from a simplified 64-player polymatrix game simulation, where the graph that represents player interactions is sparse. Here, the strategy of each player informs the RGB value of a corresponding pixel on a grid. If the system exhibits Poincaré recurrence, we should eventually see similar patterns emerge as the pixels change color over time (i.e., as their corresponding mixed strategies evolve). As observed in Figure 3, the system returns near the initial image at time $T = 6226$. Further details about the experiments can be found in Appendix F.

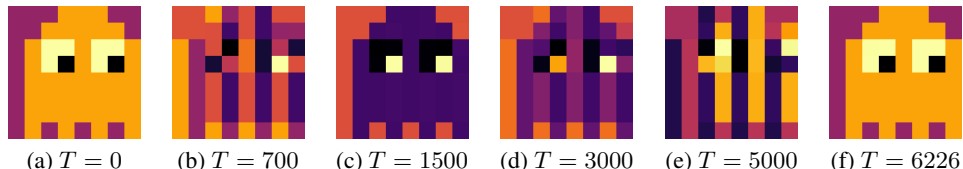

| (a) $T = 0$ | (b) $T = 700$ | (c) $T = 1500$ | (d) $T = 3000$ | (e) $T = 5000$ | (f) $T = 6226$ |

Figure 3: Sequence of images showing Poincaré recurrence in an $8 \times 8$ zero-sum polymatrix game, where the changing color of each pixel on the grid represents the mixed strategy of the player over time. After time $T = 6226$, we see that an approximation of the original image is recovered, showing that the recurrence property holds.

## 7   Discussion

We study both GDA and FTRL learning dynamics in periodically varying zero-sum games. We prove that the recurrent nature of such dynamics carries over from static games to the classes of evolving games we study. Yet, in the settings we analyze, the time-average convergence behavior from static zero-sum games can fail to generalize. This work takes a step toward understanding the behavior of classical learning algorithms for games in the more realistic setting where the game itself is not fixed.

We conclude by discussing related works on learning dynamics in evolving games. The existing literature considers a number of models that admit distinct results [19, 30, 9, 14]. In a class of time-evolving games where the evolution can be arbitrary [9], algorithms are designed that provide a novel type of regret guarantee called Nash equilibrium regret. In a sense these algorithms are competitive against the Nash equilibrium of the long-term-averaged payoff matrix. In an analysis of discrete-time FTRL dynamics in evolving games that are strictly/strongly monotone [14], sufficient conditions (e.g., when the evolving game stabilizes) under which the dynamics track/converge to the evolving equilibrium are derived. Unfortunately, zero-sum games do not satisfy these strong properties. Finally, in a model of endogenously evolving zero-sum games where a parametric game evolves itself adversarially towards the participating agents, a transformation that treats the game as an additional "hidden" agents allows for a reduction to a more standard static network zero-sum game has been developed [19, 30] under which both time-average convergence to equilibrium as well as Poincaré recurrence holds.

This growing literature indicates that time-varying games can exhibit distinct and often times more complex behavior than their classic static counterparts. As such, there is much potential for future work towards a better understanding of their learning dynamics.

## Acknowledgments and Disclosure of Funding

This research/project is supported in part by the National Research Foundation, Singapore under its AI Singapore Program (AISG Award No: AISG2-RP-2020-016), NRF 2018 Fellowship NRF-NRFF2018-07, NRF2019-NRF-ANR095 ALIAS grant, grant PIE-SGP-AI-2018-01, AME Programmatic Fund (Grant No. A20H6b0151) from the Agency for Science, Technology and Research (A*STAR). Tanner Fiez was supported by a National Defense Science and Engineering Graduate Fellowship.

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
