# Online Learning in Periodic Zero-Sum Games

## Supplementary Material

## Appendix Organization and Contents

The organization and contents of this appendix is as follows. Appendix A covers additional related work, which we provide a separate bibliography for at the end of the document. Following Appendix A, proofs for the theoretical results in the paper are presented in the order that they appeared. Specifically, Appendix B contains the proofs for the results presented in Section 4.1 on GDA dynamics and recurrence. This includes the proofs of Proposition 1, Lemma 1, Lemma 2, and Theorem 1. In Appendix C, we provide the proof of Proposition 2 from Section 4.2 on the time-average behavior of GDA. Appendix D contains the analysis for Section 5.1 on FTRL dynamics and recurrence, including the proofs of Lemma 3, Lemma 4, and Theorem 2. Appendix E covers the time-average behavior of FTRL dynamics as presented in Section 5.2. Specifically, the proofs of Theorem 3 and Proposition 3 are given in Appendix E. Finally, Appendix F contains additional experimental results.

## A    Additional Related Work

The discussions of related work presented in Section 1 and Section 7 focused on comparable theoretical results in static zero-sum games and studies on classes of certain evolving zero-sum games. We remark that there is also a rich literature studying evolutionary dynamics in action-dependent (endogenous) evolving games in problems strongly motivated by applications in science, economics, and sociology. Often this line of work falls under the hood of what is known as negative frequency dependent selection [17]. Negative frequency dependent selection is an evolutionary process in which the fitness of a strategy dissipates as it becomes more common. Indeed, a number of works (see, [34, 31] and the references therein) analyze evolutionary dynamics in specific formulations, typically in low-dimensional strategy spaces, and characterize the outcomes. In contrast, we study a broad class of learning dynamics in a general class of exogenously evolving games.

## B    GDA Recurrence Results: Proofs for Section 4.1

This appendix includes the proofs of Proposition 1, Lemma 1, Lemma 2, and Theorem 1.

### B.1    Proof of Proposition 1

To prove this result, we construct time-evolving zero-sum games without both periodic payoffs and a time-invariant equilibrium in which the GDA dynamics are not Poincaré recurrent.

**Example 1.** Consider a time-evolving zero-sum game on scalar action spaces so that $x_1, x_2 \in \mathbb{R}$ with the time-evolving payoff matrix $A(t) = t^{-2}$. Without loss of generality, we can consider $t > 0$ so that the GDA dynamics are well-defined. This time-evolving zero-sum does not have periodic payoffs, but $(x_1^*, x_2^*) = (0, 0)$ is a time-invariant Nash equilibrium. We now show that the GDA dynamics are not Poincaré recurrent in this game. Before formally proving this, we remark that the intuition for why this statement holds is that since the payoff matrix goes to zero, the distance the dynamics can travel is bounded so it is impossible that the trajectory could return arbitrarily close to an initial condition infinitely often.

The GDA dynamics in this time-evolving zero-sum game are described by the system

$$\begin{bmatrix} \dot{x}_1 \\ \dot{x}_2 \end{bmatrix} = \begin{bmatrix} 0 & \frac{1}{t^2} \\ -\frac{1}{t^2} & 0 \end{bmatrix} \begin{bmatrix} x_1(t) \\ x_2(t) \end{bmatrix}.$$

The solution of a time-varying linear system of this form is given by

$$\begin{bmatrix} x_1(t) \\ x_2(t) \end{bmatrix} = \exp\left( \int_{t_0}^{t} \begin{bmatrix} 0 & \frac{1}{\tau^2} \\ -\frac{1}{\tau^2} & 0 \end{bmatrix} d\tau \right) \begin{bmatrix} x_1(t_0) \\ x_2(t_0) \end{bmatrix}.$$

We consider $t_0 > 1$ without loss of generality. To derive the explicit solution, we begin by computing the integral of the evolving payoff matrix and get that

$$\int_{t_0}^{t} \begin{bmatrix} 0 & \frac{1}{\tau^2} \\ -\frac{1}{\tau^2} & 0 \end{bmatrix} d\tau = \begin{bmatrix} 0 & \frac{1}{t_0} - \frac{1}{t} \\ \frac{1}{t} - \frac{1}{t_0} & 0 \end{bmatrix}.$$

Recalling the following identity

$$\exp\left( \theta \begin{bmatrix} 0 & -1 \\ 1 & 0 \end{bmatrix} \right) = \begin{bmatrix} \cos(\theta) & \sin(\theta) \\ -\sin(\theta) & \cos(\theta) \end{bmatrix},$$

we can determine that the matrix exponential is then given by

$$\exp\left( \begin{bmatrix} 0 & \frac{1}{t_0} - \frac{1}{t} \\ \frac{1}{t} - \frac{1}{t_0} & 0 \end{bmatrix} \right) = \begin{bmatrix} \cos(\frac{1}{t} - \frac{1}{t_0}) & -\sin(\frac{1}{t} - \frac{1}{t_0}) \\ \sin(\frac{1}{t} - \frac{1}{t_0}) & \cos(\frac{1}{t} - \frac{1}{t_0}) \end{bmatrix}.$$

Therefore, the solution of the system simplifies to be given by

$$\begin{bmatrix} x_1(t) \\ x_2(t) \end{bmatrix} = \begin{bmatrix} \cos(\frac{1}{t} - \frac{1}{t_0}) & -\sin(\frac{1}{t} - \frac{1}{t_0}) \\ \sin(\frac{1}{t} - \frac{1}{t_0}) & \cos(\frac{1}{t} - \frac{1}{t_0}) \end{bmatrix} \begin{bmatrix} x_1(t_0) \\ x_2(t_0) \end{bmatrix},$$

and equivalently,

$$\begin{bmatrix} x_1(t) \\ x_2(t) \end{bmatrix} = \begin{bmatrix} \cos(\frac{1}{t} - \frac{1}{t_0})x_1(t_0) - \sin(\frac{1}{t} - \frac{1}{t_0})x_2(t_0) \\ \sin(\frac{1}{t} - \frac{1}{t_0})x_1(t_0) + \cos(\frac{1}{t} - \frac{1}{t_0})x_2(t_0) \end{bmatrix}.$$

Given the explicit form of the solution, we now show that we can construct an initial condition that the strategies do not return back to infinitely often. This will immediately allow us to conclude the system is not Poincaré recurrent by definition. We remark that the following choice of initial condition is only for the simplicity of the proof and identical conclusions would hold for almost all initial conditions.

Let $x_1(t_0) = 1$ and $x_2(t_0) = 0$. Given this initial condition, the solution of the system simplifies to be given by

$$\begin{bmatrix} x_1(t) \\ x_2(t) \end{bmatrix} = \begin{bmatrix} \cos(\frac{1}{t} - \frac{1}{t_0}) \\ \sin(\frac{1}{t} - \frac{1}{t_0}) \end{bmatrix}.$$

Taking the limit of the solution as $t \to \infty$, we have that

$$\lim_{t \to \infty} x_1(t) = \lim_{t \to \infty} \cos\left( \frac{1}{t} - \frac{1}{t_0} \right) = \cos\left( \frac{1}{t_0} \right)$$

and

$$\lim_{t \to \infty} x_2(t) = \lim_{t \to \infty} \sin\left( \frac{1}{t} - \frac{1}{t_0} \right) = -\sin\left( \frac{1}{t_0} \right).$$

This shows that the dynamics converge to a fixed point $(\bar{x}_1, \bar{x}_2) = (\cos\left(\frac{1}{t_0}\right), -\sin\left(\frac{1}{t_0}\right))$. Since $(\bar{x}_1, \bar{x}_2) = (\cos\left(\frac{1}{t_0}\right), \sin\left(\frac{1}{t_0}\right)) \neq (1, 0) = (x_1(t_0), x_2(t_0))$ for $t_0 > 1$ unless $t_0 \to \infty$, this immediately implies that the GDA dynamics are not Poincaré recurrent in this time-evolving zero-sum game since they do not return infinitely often back to an arbitrarily small neighborhood around the initial condition.

**Example 2.** In the previous example, we showed that the GDA dynamics were not Poincaré recurrent in a time-evolving zero-sum game that had a time-invariant Nash equilibrium but not periodic payoffs. In this example, we provide theoretical evidence that the GDA dynamics are not Poincaré recurrent in a time-evolving zero-sum game with periodic payoffs but without a time-invariant Nash equilibrium.

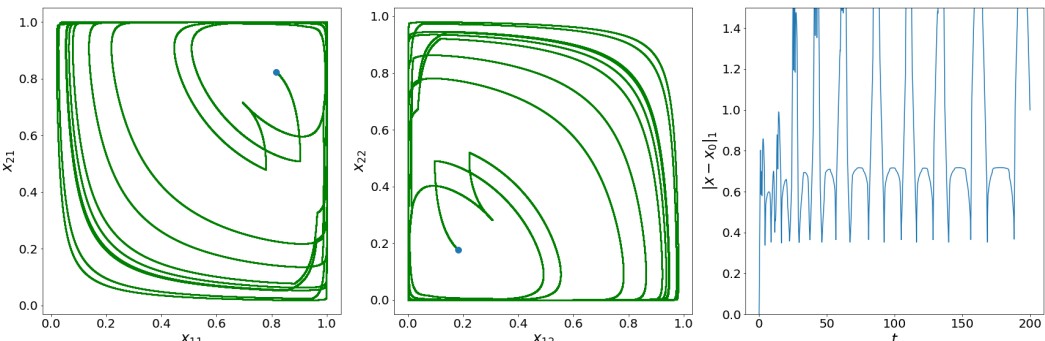

Figure 4: (Left, Center) Replicator trajectories for periodically evolving game without time-invariant equilibrium. (Right) $L_1$-norm plot showing that recurrence does not hold in this example.

Consider a time-evolving zero-sum game on scalar action spaces so that $x_1, x_2 \in \mathbb{R}$ with a periodic payoff matrix $A(t) = A(t + T)$ for any $t \geq 0$ and $T = 3$ that evolves over a period such that $A(t) = 1$ for $0 \leq t \leq 1$ and $A(t) = -1$ for $1 \leq t \leq 3$. We treat player 2 as a dummy player that just plays the fixed strategy of $x_2 = 1$ for all $t$. Thus, this time-evolving zero-sum game can be viewed as a trivial game that is equivalent to an optimization problem for player 1. Since player 1 is a utility maximizer, the Nash equilibrium of the game at each time simply corresponds to the strategy of player 1 that maximizes its utility. Thus, the Nash equilibrium when $A(t) = 1$ is $x_1^* = \infty$ and given $A(t) = -1$ it is $x_2 1^* = -\infty$. Therefore, this corresponds to a time-evolving zero-sum game that is periodic but there is not a time-invariant Nash equilibrium.

We now show that the GDA dynamics are not Poincaré recurrent in this game. The dynamics can be described by the system $\dot{x}_1 = A(t)$. Consequently, the solution is $x_1(t) = x_1(t_0) + t$ on the interval with $A(t) = 1$ and $x_1(t) = x_1(t_0) - 1$ on the interval with $A(t) = -1$. This means that after 1 period of the game, $x_1(t) = x_1(t_0) - 1$, which implies that $x_1(t) \to -\infty$ as $t \to \infty$. Thus the dynamics are not Poincaré recurrent in this time-evolving zero-sum game since they do not return infinitely often back to an arbitrarily small neighborhood around the initial condition.

**Example 3.** Additionally, we provide an experimental example to show the non-existence of Poincaré recurrence in the setting of FTRL dynamics. In particular, we simulate a time-evolving zero-sum game which has periodic payoffs but does not have a time-invariant Nash equilibrium. Consider a time-evolving zero-sum game where the payoff matrix for the first quarter of a period is standard Matching Pennies $A = \begin{pmatrix} 1 & -1 \\ -1 & 1 \end{pmatrix}$. For the remaining three quarters of the period it is instead $A = \begin{pmatrix} 0.05 & -0.5 \\ -0.5 & 5 \end{pmatrix}$. Note that here the payoff matrix for the second player is just $-A$. The former game has a mixed Nash equilibrium where both players play $[0.5, 0.5]$ and the latter game has a mixed Nash equilibrium where both players play $[0.9091, 0.0909]$. We simulate this example with replicator dynamics, which is an instantiation of FTRL dynamics. First, we plot the trajectories of the player's strategies against each other. Moreover, we plot the $L_1$-norm between the joint trajectory of the players and the initial condition. The simulation results show that the trajectory does not return back arbitrarily close to the initial condition, thus the dynamics are not Poincaré recurrent in this periodic evolving game without a time-invariant equilibrium (Figure 4).

### B.2 Proof of Lemma 1

We can show that the GDA dynamics are volume preserving by showing that the vector field is divergence free and then applying Liouville's theorem. Indeed,

$$\mathrm{div}(\dot{x}) = \sum_{i=1}^{2} \sum_{j=1}^{n_i} \frac{\partial \dot{x}_{ij}}{\partial x_{ij}} = 0,$$

which follows from the fact that $\dot{x}_{ij}$ is independent of $x_{ij}$ for each $i, j$. The divergence free property of the vector field then ensures that the flow $\phi^t$ of the differential equation is volume preserving by Liouville's theorem.

### B.3 Proof of Lemma 2

To prove this statement, we claim the following function is time-invariant:

$$\Phi(t) = \frac{1}{2}\left(x_1^\top(t)x_1(t) + x_2^\top(t)x_2(t)\right).$$

By taking the time-derivative of the $\Phi(t)$ we can verify the function is a constant of motion. Indeed, this holds based on the following analysis:

$$\begin{aligned}
\frac{d\Phi}{dt} &= \frac{1}{2}\left(x_1^\top(t)\dot{x}_1 + \dot{x}_1^\top x_1(t) + x_2^\top(t)\dot{x}_2 + \dot{x}_2^\top x_2(t)\right) \\
&= x_1^\top(t)\dot{x}_1 + x_2^\top(t)\dot{x}_2 \\
&= x_1^\top(t)A(t)x_2(t) - x_2(t)^\top A^\top(t)x_1(t) \\
&= 0.
\end{aligned}$$

Finally, observe that given a bounded initial condition, the time-invariance of $\Phi(t)$ directly implies that no strategy of any player can become unbounded so the flow $\phi^t$ of the differential equation has bounded orbits.

### B.4 Proof of Theorem 1

Given the previous intermediate results, Theorem 1 follows from the arguments presented in Section 3.3. In particular, observe that by definition of the periodic zero-sum bilinear game, the GDA dynamics are $T$-periodic. Now, consider the discrete-time dynamical system defined by the Poincaré map $\phi^T$ that arises. This system retains the volume preservation property of the continuous-time system from Lemma 1 since as presented in Section 3.3, if a $T$-periodic system is divergence-free then the discrete-time system defined by $\phi^T$ is also volume preserving [1, 3.16.B, Thm 2]. Similarly, the discrete-time system defined by the $\phi^T$ retains the bounded orbits guarantee of the continuous-time system from Lemma 2 since it holds at any set of times. Thus, we are able to apply the Poincaré recurrence theorem for discrete-time systems from Section 3.3 to the discrete-time system defined by $\phi^T$ to conclude the discrete-time system is Poincaré recurrent. This immediately implies that the GDA dynamics are Poincaré recurrent since the discrete-time system defined by $\phi^T$ forms a subsequence of the continuous-time system.

## C GDA Time-Average Result: Proof of Proposition 2

Consider a periodic zero-sum bilinear game with $x_1, x_2 \in \mathbb{R}$ and a periodic payoff matrix $A(t)$ such that $A(t) = A(t + T)$ with $T = 3\pi$ for any $t \geq 0$. Moreover, let the payoff matrix evolve over a period as follows:

$$A(t) = \begin{cases} -1 & 0 \leq t \leq \pi \\ 1 & \pi \leq t \leq \frac{3\pi}{2} \\ -1 & \frac{3\pi}{2} \leq t \leq 3\pi. \end{cases}$$

The joint strategy $(x_1^*, x_2^*) = (0, 0)$ is the time-invariant Nash equilibrium. We now show that the time-average of the strategies produced by the GDA dynamics do not converge to the time-invariant Nash equilibrium.

The GDA dynamics in this periodic zero-sum bilinear game are given by

$$\begin{aligned}
\dot{x}_1 &= A(t)x_2(t) \\
\dot{x}_2 &= -A(t)x_1(t).
\end{aligned}$$

The solution to the differential equation that describes the GDA dynamics can be constructed in a piecewise manner. On each of the three intervals we have a linear system defined by

$$\begin{bmatrix} \dot{x}_1 \\ \dot{x}_2 \end{bmatrix} = \begin{bmatrix} 0 & A(t) \\ -A(t) & 0 \end{bmatrix} \begin{bmatrix} x_1 \\ x_2 \end{bmatrix}.$$

Now recall the following identity

$$\exp\left(\theta \begin{bmatrix} 0 & 1 \\ -1 & 0 \end{bmatrix}\right) = \begin{bmatrix} \cos(\theta) & \sin(\theta) \\ -\sin(\theta) & \cos(\theta) \end{bmatrix}.$$

Hence for initial condition $(x_1(0), x_2(0))$ and interval $[0, \pi)$ we know that $A(t) = -1$ for all $t$ in the interval which implies that the solution on this interval is given by

$$\begin{bmatrix} x_1(t) \\ x_2(t) \end{bmatrix} = \begin{bmatrix} \cos(-t) & \sin(-t) \\ -\sin(-t) & \cos(-t) \end{bmatrix} \begin{bmatrix} x_1(0) \\ x_2(0) \end{bmatrix}.$$

On the interval $[\pi, 3\pi/2)$, $A(t) = 1$ so that

$$\begin{bmatrix} x_1(t) \\ x_2(t) \end{bmatrix} = \begin{bmatrix} \cos(t - \pi) & \sin(t - \pi) \\ -\sin(t - \pi) & \cos(t - \pi) \end{bmatrix} \begin{bmatrix} x_1(\pi) \\ x_2(\pi) \end{bmatrix}.$$

Finally, on $[3\pi/2, 3\pi)$, $A(t) = -1$ so that

$$\begin{bmatrix} x_1(t) \\ x_2(t) \end{bmatrix} = \begin{bmatrix} \cos(-(t - 3\pi/2)) & \sin(-(t - 3\pi/2)) \\ -\sin(-(t - 3\pi/2)) & \cos(-(t - 3\pi/2)) \end{bmatrix} \begin{bmatrix} x_1(3\pi/2) \\ x_2(3\pi/2) \end{bmatrix}.$$

Now, let us consider the initial condition $(x_1(0), x_2(0)) = (1, 0)$. Then,

$$(x_1(t), x_2(t)) = \begin{cases} (\cos(t), \sin(t)) = (\cos(-t), -\sin(-t)) & t \in [0, \pi) \\ (\cos(t), -\sin(t)) = (-\cos(t - \pi), \sin(t - \pi)) & t \in [\pi, 3\pi/2) \\ (-\cos(t), -\sin(t)) = (\sin(-(t - 3\pi/2)), \cos(-(t - 3\pi/2))) & t \in [3\pi/2, 3\pi) \end{cases}$$

Observe that from this solution we can determine that the GDA dynamics return to the initial condition at the end of a period. Thus, to assess convergence of the time-average it is sufficient to evaluate the time-average of the dynamics over a period of the evolving game. Integrating the solution over a period, we have that

$$\int_0^{3\pi} x_1(t)dt = \int_0^{3\pi/2} \cos(t)dt + \int_{3\pi/2}^{3\pi} -\cos(t)dt$$
$$= [\sin(3\pi/2) - \sin(0)] - [\sin(3\pi) - \sin(3\pi/2)]$$
$$= -2$$

and

$$\int_0^{3\pi} x_2(t)dt = \int_0^{\pi} \sin(t)dt + \int_{\pi}^{3\pi} -\sin(t)dt$$
$$= [-\cos(\pi) + \cos(0)] + [\cos(3\pi) - \cos(\pi)]$$
$$= 2$$

This implies that the time-average strategies of the players do not equal to zero, so the time-average of the GDA dynamics do not converge to the time-invariant Nash equilibrium. The fact that it is non-zero holds generally even changing the initial condition.

This completes the proof and shows that there exists periodic zero-sum bilinear games where the time-average GDA strategies do not converge to the time-invariant Nash equilibrium.

## D    FTRL **Poincaré Recurrence Results: Proofs for Section 5.1**

This appendix includes the proofs of Lemma 3, Lemma 4, and Theorem 2.

### D.1    Proof of Lemma 3

Recall that this result states that the dynamics defined by the system $\dot{z}$ (given again in (3)) are volume preserving in any periodic zero-sum polymatrix game. Here, we also explain how the $\dot{z}$ dynamics were formulated. Then, we show that the divergence of this vector field is zero, from which we conclude the dynamics are volume preserving by Liouville's theorem. This proof follows closely arguments in [20].

For each player $i \in V$, given a fixed strategy $\beta \in \mathcal{A}_i$, for all $\alpha \in \mathcal{A}_i \setminus \beta$ the cumulative utility differences are defined by
$$z_{i\alpha}(t) = y_{i\alpha}(t) - y_{i\beta}(t).$$

This transformation from the cumulative utilities to the cumulative utility differences yields a linear map $\Pi_i : \mathbb{R}^{n_i} \to \mathbb{R}^{n_i-1}$ from $y_i(t)$ to $z_i(t)$ for each player $i \in V$. Moreover, define by $\Pi = (\Pi_1, \ldots, \Pi_{|V|})$ the product map of the linear maps $\Pi_i$ of each player $i \in V$. This map is surjective, but not injective.

Observe that the cumulative utility differences for each player $i \in V$ and all $\alpha \in \mathcal{A}_i \setminus \beta$ evolve following the differential equation

$$\dot{z}_{i\alpha}(t) = v_{i\alpha}(x(t), t) - v_{i\beta}(x(t), t). \tag{3}$$

The above differential equation is obtained directly by the form of $y_i$ and the fundamental theorem of calculus. Moreover, recall that for any player $i \in V$ and pure strategy $\gamma \in \mathcal{A}_i$, the quantity $v_{i\gamma}(x(t), t)$ gives utility of player $i \in V$ at any time $t \geq 0$ for selecting the pure strategy $\gamma \in \mathcal{A}_i$.

To analyze the dynamics from the system in (3) we need it to be well-defined, which is not immediate since it depends on $x(t) = Q(y(t))$ and the mapping from $y(t)$ to $z(t)$ via $\Pi$ is not invertible so that $y(t)$ cannot be expressed as a fuction of $z(t)$. However, despite this, the system is in fact well-defined. To see this, for each player $i \in V$, consider the reduced choice map $\hat{Q}_i : \mathbb{R}^{n_i-1} \to \mathcal{X}_i$ defined as $\hat{Q}_i(z_i(t)) = Q_i(y_i(t))$ for some $y_i(t) \in \mathbb{R}^{n_i}$ such that $\Pi_i(y_i(t)) = z_i(t)$ which is guaranteed to exist since $\Pi_i$ is surjective. Then, the fact that $\hat{Q}_i(z_i(t))$ is well-defined for each player $i \in V$ holds since by the construction $\Pi_i(y_i(t)) = \Pi_i(y'_i(t))$ if and only if $y'_{i\alpha}(t) = y_{i\alpha}(t) + c$ for $c \in \mathbb{R}$ and every $\alpha_i \in \mathcal{A}_i$ which immediately implies $Q_i(y'_i(t)) = Q_i(y_i(t))$ if and only if $\Pi_i(y_i(t)) = \Pi_i(y'_i(t))$. Finally, let $\hat{Q} = (\hat{Q}_1, \ldots, \hat{Q}_{|V|})$ be the combined reduced choice map and note that $Q(y(t)) = \hat{Q}(\Pi(y(t))) = \hat{Q}(z(t))$ by the construction. As a result, the dynamics from the system in (3) are equivalently given by the following system

$$\dot{z}_{i\alpha} = v_{i\alpha_i}(\hat{Q}_i(z(t), t)) - v_{i\beta_i}(\hat{Q}_i(z(t)), t).$$

This system is well-defined by the arguments above which ensures that the system in (3) is well-defined.

Now that we have shown the system is well-defined, we prove that is is volume preserving. To see this, observe that the vector field is divergence free. Indeed,

$$\mathrm{div}(\dot{z}) = \sum_{i \in V} \sum_{\alpha \in \mathcal{A}_i} \frac{\partial \dot{z}_{i\alpha}}{\partial z_{i\alpha}} = \sum_{i \in V} \sum_{\alpha \in \mathcal{A}_i} \sum_{\gamma_i \in \mathcal{A}_i} \frac{\partial \dot{z}_{i\alpha}}{\partial x_{i\gamma}} \frac{\partial x_{i\gamma}}{\partial z_{i\alpha_i}} = 0.$$

Note that the equation above holds since for each player $i \in V$, the pure strategy utilities at any time $t \geq 0$ given by $v_i(x(t), t)$ where $v_{i\alpha}(x(t), t) = u_i((\alpha, x_{-i}(t)), t)$ do not depend on $x_i(t)$. Finally, the divergence free property of the vector field ensures that the flow $\phi^t$ of the differential equation is volume preserving by Liouville's theorem.

## D.2 Proof of Lemma 4

Recall that Lemma 4 states that the orbits of the $\dot{z}$ dynamics are bounded. To prove this statement, we show that the function

$$\Phi(x^*, y(t)) = \sum_{i \in V} \left( h_i^*(y_i(t)) - \langle x_i^*, y_i(t) \rangle + h_i(x_i^*) \right)$$

is time-invariant where $x^*$ denotes the time-invariant fully mixed Nash equilibrium and then argue that this is sufficient to ensure that orbits of the $\dot{z}$ dynamics are bounded.

To prove that the function $\Phi(x^*, y(t))$ is time-invariant, we show that the time-derivative of the function is equal to zero. The time-derivative of $\Phi(x^*, y(t))$ simplifies using the fact that $h_i(x_i^*)$ is time-independent to the following:

$$\frac{d\Phi(x^*, y(t))}{dt} = \frac{d}{dt} \sum_{i \in V} h_i^*(y_i(t)) + \frac{d}{dt} \sum_{i \in V} \langle x_i^*, y_i(t) \rangle.$$

We begin by showing that the time-derivative of $\sum_{i \in V} h_i^*(y_i(t)) = 0$. This holds by the following computation that is explained below:

$$\frac{d}{dt} \sum_{i \in V} h_i^*(y_i(t)) = \sum_{i \in V} \langle \nabla h_i^*(y_i(t)), \dot{y}_i(t) \rangle \tag{4}$$

$$= \sum_{i \in V} \langle x_i(t), \dot{y}_i(t) \rangle \tag{5}$$

$$= \sum_{i \in V} \langle x_i(t), v_i(x(t), t) \rangle \tag{6}$$

$$= \sum_{i \in V} u_i(x(t), t) \tag{7}$$

$$= 0. \tag{8}$$

We obtain (4) by the chain rule, (5) by the maximizing argument of convex conjugates (see e.g., Shalev-Shwartz et al. 29, Chapter 2) that implies $x_i(t) = Q_i(y_i(t)) = \nabla h_i^*(y_i(t))$, (6) by the definition of $y_i(t)$ and the fundamental theorem of calculus, (7) by definition of the pure strategy utilities $v_i(x(t), t)$ and the utility $u_i(x(t), t)$, and (8) by the fact that the polymatrix game is zero-sum.

We now finish by showing that the time-derivative of $\sum_{i \in V} \langle x_i^*, y_i(t) \rangle = 0$. To begin, observe that the time-derivative can be described by

$$\frac{d}{dt} \sum_{i \in V} \langle x_i^*, y_i(t) \rangle = \sum_{i \in V} \langle x_i^*, \dot{y}_i(t) \rangle$$

$$= \sum_{i \in V} \sum_{j:(i,j) \in E} (x_i^*)^\top A^{ij}(t) x_j(t) \tag{9}$$

$$= \sum_{i \in V} \sum_{j:(i,j) \in E} (x_i^*)^\top A^{ij}(t)(x_j(t) - x_j^*). \tag{10}$$

Observe that (9) follows from the definition of $y_i(t)$ and the fundamental theorem of calculus and (10) comes about from subtracting $\sum_{i \in V} u_i(x^*, t)$ which is zero by the fact that the polymatrix game is zero-sum for any $t \geq 0$.

To continue, we remark that any zero-sum polymatrix game can be transformed to a payoff equivalent, pairwise constant-sum game [8]. This means that for each edge $(i, j) \in E$ there exists a matrix $B^{ij}(t)$ such that the following properties hold (see Lemma 3.1, 3.2, and 3.4, respectively Cai and Daskalakis 8):

**Property 1.** $A_{\alpha\beta}^{ij}(t) - A_{\alpha\gamma}^{ij}(t) = B_{\alpha\beta}^{ij}(t) - B_{\alpha\gamma}^{ij}(t)$ for any pure strategies $\alpha \in \mathcal{A}_i$ and $\beta, \gamma \in \mathcal{A}_j$.

**Property 2.** $B^{ij}(t) + (B^{ji}(t))^\top = c_{ij}(t) \cdot \mathbf{1}_{n_i \times n_j}$, where $c_{ij}(t)$ is a constant and $\mathbf{1}_{n_i \times n_j}$ is an $n_i \times n_j$ matrix of ones.

**Property 3.** In every joint pure strategy profile, every player $i \in V$ has the same utility in the game defined by the individual payoff matrices $\{A^{ij}(t)\}_{(i,j) \in E}$ as in the game defined by the individual payoff matrices $\{B^{ij}(t)\}_{(i,j) \in E}$.

Fixing a strategy $\gamma \in \mathcal{A}_j$, we can equivalently express any summand of (10) in the following manner that is justified below:

$$(x_i^*)^\top A^{ij}(x_j(t) - x_j^*) = \sum_{\alpha \in \mathcal{A}_i} \sum_{\beta \in \mathcal{A}_j} x_{i\alpha}^* A_{\alpha\beta}^{ij}(x_{j\beta}(t) - x_{j\beta}^*)$$

$$= \sum_{\alpha \in \mathcal{A}_i} \sum_{\beta \in \mathcal{A}_j} x_{i\alpha}^* \left( B_{\alpha\beta}^{ij}(t) - B_{\alpha\gamma}^{ij}(t) + A_{\alpha\gamma}^{ij}(t) \right)(x_{j\beta}(t) - x_{j\beta}^*) \tag{11}$$

$$= (x_i^*)^\top B^{ij}(t)(x_j(t) - x_j^*) + \sum_{\alpha \in \mathcal{A}_i} x_{i\alpha}^* \left( A_{\alpha\gamma}^{ij}(t) - B_{\alpha\gamma}^{ij}(t) \right) \sum_{\beta \in \mathcal{A}_j} (x_{j\beta}(t) - x_{j\beta}^*).$$

$$= (x_i^*)^\top B^{ij}(t)(x_j(t) - x_j^*). \tag{12}$$

Observe that (11) results from applying Property 1 and (12) holds since both $x_j(t)$ and $x_j^*$ are on the simplex so that $\sum_{\beta \in \mathcal{A}_j} x_{j\beta} = 1$ and $\sum_{\beta \in \mathcal{A}_j} x_{j\beta}^* = 1$ which implies $\sum_{\beta \in \mathcal{A}_j}(x_{j\beta} - x_{j\beta}^*) = 0$.

Thus, continuing from (10) and using that $(x_i^*)^\top A^{ij}(x_j(t) - x_j^*) = (x_i^*)^\top B^{ij}(t)(x_j(t) - x_j^*)$ from above and swapping the sum indexing and taking the transpose of the quadratic form $(x_i^*)^\top B^{ij}(t)(x_j(t) - x_j^*)$, we get that

$$\frac{d}{dt} \sum_{i \in V} \langle x_i^*, y_i(t) \rangle = \sum_{i \in V} \sum_{j:(i,j) \in E} (x_i^*)^\top A^{ij}(t)(x_j(t) - x_j^*)$$

$$= \sum_{i \in V} \sum_{j:(i,j) \in E} (x_i^*)^\top B^{ij}(t)(x_j(t) - x_j^*) \tag{13}$$

$$= \sum_{j \in V} \sum_{i:(j,i) \in E} (x_j(t) - x_j^*)^\top (B^{ij}(t))^\top x_i^*.$$

Moreover, we obtain the following expression that is justified below:

$$\frac{d}{dt} \sum_{i \in V} \langle x_i^*, y_i(t) \rangle = \sum_{j \in V} \sum_{i:(j,i) \in E} (x_j(t) - x_j^*)^\top (B^{ij}(t))^\top x_i^*$$

$$= \sum_{j \in V} \sum_{i:(j,i) \in E} (x_j(t) - x_j^*)^\top (c^{ji}(t) \mathbf{1}_{n_j \times n_i} - B^{ji}(t)) x_i^* \tag{14}$$

$$= \sum_{j \in V} \sum_{i:(j,i) \in E} c^{ji}(t)(x_j(t) - x_j^*)^\top \mathbf{1}_{n_j \times n_i} x_i^* - \sum_{j \in V} \sum_{i:(j,i) \in E} (x_j(t) - x_j^*)^\top B^{ji}(t) x_i^*$$

$$= -\sum_{j \in V} \sum_{i:(j,i) \in E} (x_j(t) - x_j^*)^\top B^{ji}(t) x_i^*. \tag{15}$$

Note that (14) results from applying Property 2 and we obtain (15) using that $\sum_{j \in V} \sum_{i:(j,i) \in E} c^{ji}(t)(x_j(t) - x_j^*)^\top = 0$ since each summand is zero as can be seen by noting that $\sum_{\alpha \in \mathcal{A}_j} x_{j\alpha} = \sum_{\alpha \in \mathcal{A}_j} x_{j\alpha}^* = \sum_{\alpha \in \mathcal{A}_i} x_{i\alpha}^* = 1$ which gives

$$c^{ji}(t)(x_j(t) - x_j^*)^\top \mathbf{1}_{n_j \times n_i} x_i^* = c^{ji}(x_j(t) - x_j^*)^\top \mathbf{1}_{n_j} = c^{ji}(t) - c^{ji}(t) = 0.$$

We now analyze the summand in (15) for some $j \in V$. Fixing any pure strategy $\gamma_i \in \mathcal{A}_i$ for each $i \in V \setminus \{j\}$, obtain the following simplification that is explained below:

$$\sum_{i:(j,i) \in E} (x_j(t) - x_j^*)^\top B^{ji}(t) x_i^* = \sum_{i:(j,i) \in E} \sum_{\alpha \in \mathcal{A}_j} \sum_{\beta \in \mathcal{A}_i} (x_{j\alpha}(t) - x_{j\alpha}^*) B_{\alpha\beta}^{ji}(t) x_{i\beta}^*$$

$$= \sum_{i:(j,i) \in E} \sum_{\alpha \in \mathcal{A}_j} \sum_{\beta \in \mathcal{A}_i} (x_{j\alpha}(t) - x_{j\alpha}^*) \big(A_{\alpha\beta}^{ji}(t) - A_{\alpha\gamma_i}^{ji}(t) + B_{\alpha\gamma_i}^{ji}(t)\big) x_{i\beta}^* \tag{16}$$

$$= \sum_{i:(j,i) \in E} (x_j(t) - x_j^*)^\top A^{ji}(t) x_i^* + \sum_{\alpha \in \mathcal{A}_j} (x_{j\alpha}(t) - x_{j\alpha}^*) \sum_{i:(j,i) \in E} (B_{\alpha\gamma_i}^{ji}(t) - A_{\alpha\gamma_i}^{ji}(t)) \sum_{\beta \in \mathcal{A}_i} x_{i\beta}^*$$

$$= \sum_{i:(j,i) \in E} (x_j(t) - x_j^*)^\top A^{ji}(t) x_i^* + \sum_{\alpha \in \mathcal{A}_j} (x_{j\alpha}(t) - x_{j\alpha}^*) \sum_{i:(j,i) \in E} (B_{\alpha\gamma_i}^{ji}(t) - A_{\alpha\gamma_i}^{ji}(t)) \tag{17}$$

$$= \sum_{i:(j,i) \in E} (x_j(t) - x_j^*)^\top A^{ji}(t) x_i^*. \tag{18}$$

The equation in (16) follows from applying Property 1 and the equation in (17) holds since $\sum_{\beta \in \mathcal{A}_i} x_{i\beta}^* = 1$ as a result of the strategy spaces being on the simplex. Finally, to see how (18) is obtained, observe that for each $\alpha \in \mathcal{A}_j$ the terms $\sum_{i:(j,i) \in E} A_{\alpha\gamma_i}^{ji}(t)$ and $\sum_{i:(j,i) \in E} B_{\alpha\gamma_i}^{ji}(t)$ give the utility of player $j \in V$ in the games with payoffs $\{A^{ji}(t)\}_{(j,i) \in E}$ and $\{B^{ji}(t)\}_{(j,i) \in E}$ respectively under a joint pure strategy. Hence, by Property 3, the respective utilities are equal so that the difference is zero.

Finally, relating (18) back to (15), we conclude that the time-derivative is zero:

$$\frac{d}{dt} \sum_{i \in V} \langle x_i^*, y_i(t) \rangle = - \sum_{j \in V} \sum_{i:(j,i) \in E} (x_j(t) - x_j^*)^\top B^{ji}(t) x_i^*$$

$$= - \sum_{j \in V} \sum_{i:(j,i) \in E} (x_j(t) - x_j^*)^\top A^{ji}(t) x_i^* = 0.$$

The final equality holds since $x^*$ is an interior Nash equilibrium, which implies $u_{j\alpha}(x^*, t) = u_j(x^*, t)$ for all strategies $\alpha \in \mathcal{A}_j$ and any linear combination thereof.

Hence,

$$\frac{d\Phi(x^*, y(t))}{dt} = \frac{d}{dt} \sum_{i \in V} h_i^*(y_i(t)) + \frac{d}{dt} \sum_{i \in V} \langle x_i^*, y_i(t) \rangle = 0,$$

which implies that $\Phi(x^*, y(t))$ is time-invariant.

Finally, by Lemma D.2 of Mertikopoulos et al. [20], the time-invariance of $\Phi$ is sufficient to ensure that the flow $\phi^t$ of the differential equation $\dot{z}$ has bounded orbits. This finishes the proof.

### D.3 Proof of Theorem 2

Theorem 2 states that the FTRL dynamics are Poincaré recurrent in periodic zero-sum polymatrix games. The proof of this claim follows from Lemma 3, Lemma 4 and the methods described in Section 3.3. Indeed, to begin, observe that the $\dot{z}$ dynamics given in (3) are $T$-periodic. This follows immediately from the definition of a $T$-periodic system as described in Section 3.3 and the fact that the payoff matrices are $T$-periodic. Consider the discrete-time dynamical system defined by the Poincaré map $\phi^T$ that arises. This system retains the volume preservation property of the continuous-time system from Lemma 3 since as presented in Section 3.3, if a $T$-periodic system is divergence-free then the discrete-time system defined by $\phi^T$ is also volume preserving [1, 3.16.B, Thm 2]. Furthermore, the bounded orbits guarantee of the continuous-time system from Lemma 4 imply the discrete-time system defined by $\phi^T$ has bounded orbits since it is a subsequence of the continuous-time system.

Thus, we are able to apply the Poincaré recurrence theorem to the system defined by $\phi^T$ to conclude that the discrete-time system is Poincaré recurrent. This implies that the $\dot{z}$ dynamics are Poincaré recurrent since the discrete-time system defined by $\phi^T$ forms a subsequence of the continuous-time system. Finally, the Poincaré recurrence of the $\dot{z}$ dynamics directly imply the Poincaré recurrence of the FTRL strategies. Indeed, since there is an increasing sequence of times $t_n$ such that $z(t_n) \to z(0)$ by Poincaré recurrence, so using continuity there is also an increasing sequence of times $t_n$ such that $x(t_n) = Q(y(t_n)) = \hat{Q}(z(t_n)) \to \hat{Q}(z(t_0)) = x(0)$ which means the FTRL dynamics are Poincaré recurrent.

## E  FTRL Time-Average Convergence Results: Proofs for Section 5.2

This appendix includes the proofs of Theorem 3 and Proposition 3.

### E.1 Proof of Theorem 3

The outline of this proof is as follows. We begin by restating the relevant notation specialized to periodic zero-sum bimatrix games and then provide a more formal mathematical statement of the claim being proven. Following that we introduce a technical result regarding the bounded regret property of FTRL dynamics and state the implications that can be drawn from it. Finally, using the implications of bounded regret and properties of zero-sum bimatrix games we reach the conclusion.

**Notation.** Recall that we consider a periodic zero-sum bimatrix game for this result, which is a game that consists of a pair of players $i$ and $j$ and the bimatrix game between them at time $t \geq 0$ is described by the pair of payoffs $\{A^{ij}(t), A^{ji}(t)\} = \{A(t), -A^\top(t)\}$ and the sequence is periodic so that $\{A^{ij}(t+T), A^{ji}(t+T)\} = \{A^{ij}(t), A^{ji}(t)\}$ or equivalently $\{A(t+T), -A^\top(t+T)\} = \{A(t), -A^\top(t)\}$ for some finite period $T$ and all time $t \geq 0$. Moreover, the bimatrix game at each time $t \geq 0$ is zero-sum which means that $u_i(x_i, x_j, t) + u_j(x_i, x_j, t) = 0$ for any strategy pair

$x_i \in \mathcal{X}_i$ and $x_j \in \mathcal{X}_j$. Observe that we include the time-dependence $t$ in the notation of player's utility to make it explicit the utility is time-dependent as a result of the time-varying payoff matrix. Finally, let the strategy pair $(x_i^*, x_j^*) \in \mathcal{X}_i \times \mathcal{X}_j$ denote the time-invariant Nash equilibrium.

**Formal Statement of Result.** Our goal is to prove that the time-average utility of each player converges to the time-average of the values of the games over a period. That is, we seek to show

$$\lim_{t \to \infty} \frac{1}{t} \int_0^t u_i(x(\tau), \tau) d\tau = \frac{1}{T} \int_0^T u_i(x_i^*, x_j^*, \tau) d\tau = \bar{V} \tag{19}$$

and

$$\lim_{t \to \infty} \frac{1}{t} \int_0^t u_j(x(\tau), \tau) d\tau = \frac{1}{T} \int_0^T u_j(x_j^*, x_i^*, \tau) d\tau = -\bar{V}, \tag{20}$$

where $\bar{V} := \frac{1}{T} \int_0^T V(\tau) d\tau$ denotes the time-average of the values of the games over a period and $V(\tau)$ denotes the value of the game at time $\tau$ for any $\tau \geq 0$.

**Bounded Regret Property.** The proof of the above statement requires an intermediate technical result. The following result of Mertikopoulos et al. [20] states that regardless of what other players do in a polymatrix game (not necessarily zero-sum), if a player follows FTRL learning dynamics then the regret of the player is bounded. It is important to remark that this result directly applies to periodic zero-sum polymatrix games. This follows from the fact that there is no assumptions on the behavior of other players, so the dynamics from the game can be viewed as arising from the behavior of the other players.

**Proposition 4** (Theorem 3.1, Mertikopoulos et al. 20). *Let $h_{\max,i} = \max_{x_i \in \mathcal{X}_i} h_i(x_i)$ and $h_{\min,i} = \min_{x_i \in \mathcal{X}_i} h_i(x_i)$. If player $i \in V$ in a polymatrix game follows* FTRL *dynamics, then for every continuous trajectory of play $x_{-i}(t)$ of the opponents of player $i$ the following regret bound holds:*

$$\max_{x_i \in \mathcal{X}_i} \frac{1}{t} \int_0^t \left[ u_i(x_i, x_{-i}(\tau), \tau) - u_i(x(\tau), \tau) \right] d\tau \leq \frac{h_{\max,i} - h_{\min,i}}{t}.$$

**Implications of Bounded Regret.** Proposition 4 ensures that the following bounds hold for the regret of player $i$:

$$\frac{1}{t} \int_0^t \left[ u_i(x_i^*, x_j(\tau), \tau) - u_i(x(\tau), \tau) \right] d\tau \leq \max_{x_i \in \mathcal{X}_i} \frac{1}{t} \int_0^t \left[ u_i(x_i, x_j(\tau), \tau) - u_i(x(\tau), \tau) \right] d\tau$$

$$\leq \frac{h_{\max,i} - h_{\min,i}}{t}. \tag{21}$$

Similarly, Proposition 4 guarantees the following bounds hold for the regret of player $j$:

$$\frac{1}{t} \int_0^t \left[ u_j(x_j^*, x_i(\tau), \tau) - u_j(x(\tau), \tau) \right] d\tau \leq \max_{x_j \in \mathcal{X}_j} \frac{1}{t} \int_0^t \left[ u_j(x_j, x_i(\tau), \tau) - u_j(x(\tau), \tau) \right] d\tau$$

$$\leq \frac{h_{\max,j} - h_{\min,j}}{t}. \tag{22}$$

Observe that the lower bounds in (21) and (22) hold by replacing the maximizing argument over the strategy space of a player with a fixed strategy. In particular, the fixed strategy is taken to be the invariant Nash equilibrium strategy for player $i$ or $j$.

Now, taking the limit as $t \to \infty$ of each side of (21) and using the zero-sum property of the bimatrix game at each time $\tau \geq 0$, we obtain the following:

$$\lim_{t \to \infty} \frac{1}{t} \int_0^t \left[ u_i(x_i^*, x_j(\tau), \tau) - u_i(x(\tau), \tau) \right] d\tau = \lim_{t \to \infty} \frac{1}{t} \int_0^t \left[ u_i(x_i^*, x_j(\tau), \tau) + u_j(x(\tau), \tau) \right] d\tau$$

$$\leq 0. \tag{23}$$

Similarly, taking the limit as $t \to \infty$ of each side of (22) and using the zero-sum property of the bimatrix game at each time $\tau \geq 0$, we get that:

$$\lim_{t \to \infty} \frac{1}{t} \int_0^t \left[ u_j(x_j^*, x_i(\tau), \tau) - u_j(x(\tau), \tau) \right] d\tau = \lim_{t \to \infty} \frac{1}{t} \int_0^t \left[ u_j(x_j^*, x_i(\tau), \tau) + u_i(x(\tau), \tau) \right] d\tau$$

$$\leq 0. \tag{24}$$

**Time-Average Utility Convergence.** We now proceed to show that

$$\lim_{t\to\infty} \frac{1}{t} \int_0^t u_i(x_i^*, x_j^*, \tau)d\tau \le \lim_{t\to\infty} \frac{1}{t} \int_0^t u_i(x(\tau), \tau)d\tau \le \lim_{t\to\infty} \frac{1}{t} \int_0^t u_i(x_i^*, x_j^*, \tau)d\tau. \quad (25)$$

The lower bound on the time-average utility of player $i$ holds by the following analysis that is explained below:

$$\lim_{t\to\infty} \frac{1}{t} \int_0^t u_i(x(\tau), \tau)d\tau \ge \lim_{t\to\infty} \frac{1}{t} \int_0^t u_i(x_i^*, x_j(\tau), \tau)d\tau \quad (26)$$

$$\ge \lim_{t\to\infty} \frac{1}{t} \int_0^t \min_{x_j \in \mathcal{X}_j} u_i(x_i^*, x_j, \tau)d\tau \quad (27)$$

$$= \lim_{t\to\infty} \frac{1}{t} \int_0^t u_i(x_i^*, x_j^*, \tau)d\tau. \quad (28)$$

The inequality in (26) is a direct implication of (23). Moreover, the inequality in (27) is immediate by the fact that any fixed strategy of player $j$ must give at least as much utility to player $i$ as the strategy which minimizes the utility of player $i$. Finally, the last conclusion in (28) holds by the definition of a Nash equilibrium in a zero-sum bimatrix game.

The upper bound on the time-average utility of player $i$ holds by the following similar analysis that is detailed below:

$$\lim_{t\to\infty} \frac{1}{t} \int_0^t u_i(x(\tau), \tau)d\tau \le - \lim_{t\to\infty} \frac{1}{t} \int_0^t u_j(x_j^*, x_i(\tau), \tau)d\tau \quad (29)$$

$$= \lim_{t\to\infty} \frac{1}{t} \int_0^t u_i(x_i(\tau), x_j^*, \tau)d\tau \quad (30)$$

$$\le \lim_{t\to\infty} \frac{1}{t} \int_0^t \max_{x_i \in \mathcal{X}_i} u_i(x_i, x_j^*, \tau)d\tau \quad (31)$$

$$= \lim_{t\to\infty} \frac{1}{t} \int_0^t u_i(x_i^*, x_j^*, \tau)d\tau. \quad (32)$$

The inequality in (29) follows directly from (24) and the equality in (30) is a result of the zero-sum property of the game at each time $\tau \ge 0$. Furthermore, the inequality in (31) holds by the fact that the the strategy of player $i$ that maximizes the utility must give at least as much utility as any fixed strategy. The last conclusion in (32) again holds by the definition of a Nash equilibrium in a zero-sum bimatrix game.

The preceding arguments prove that the claimed inequalities in (25) hold. Observe that the time-average of the utility values of player $i$ at the invariant Nash equilibrium converge to the time-average of the values of the games over a period as a result of the periodic nature of the game and the fact that the utility value at any Nash equilibrium in a zero-sum bimatrix game is unique. That is,

$$\lim_{t\to\infty} \frac{1}{t} \int_0^t u_i(x_i^*, x_j^*, \tau)d\tau = \frac{1}{T} \int_0^T u_i(x_i^*, x_j^*, \tau)d\tau = \bar{V}.$$

Thus, the squeeze theorem applied to (25) allows us to conclude the statement given in (19):

$$\lim_{t\to\infty} \frac{1}{t} \int_0^t u_i(x(\tau), \tau)d\tau = \frac{1}{T} \int_0^T u_i(x_i^*, x_j^*, \tau)d\tau = \bar{V}$$

Finally, by the zero-sum property of the bimatrix game at each time $\tau \ge 0$, the statement given in (20) immediately follows from the equation above. That is,

$$\lim_{t\to\infty} \frac{1}{t} \int_0^t u_j(x(\tau), \tau)d\tau = \frac{1}{T} \int_0^T u_i(x_j^*, x_i^*, \tau)d\tau = -\bar{V}.$$

This finishes the proof.

## E.2 Proof of Proposition 3

We now provide the proof of Proposition 3 stating that there exists periodic zero-sum bimatrix games satisfying Definition 2 in which the time-average strategies of FTRL dynamics fail to converge to the time-invariant Nash equilibrium.

To prove this result we construct a specific periodic zero-sum bimatrix game that is the basis of the counterexample. In order to demonstrate that the FTRL strategies may not converge to the time-invariant Nash equilibrium, we consider the regularization function that leads to the replicator dynamics. Then, for replicator dynamics in the constructed game, we prove that the strategies are symmetric about the half period of the game and consequently return to the initial condition in a period of the game so that the time-average of the strategies in the limit corresponds to the time-average of the strategies over a half-period of the game. Finally, we use this property to show that the choice of the period of the game can ensure that the time-average strategies cannot converge to the time-invariant Nash equilibrium.

**Counterexample Construction.** A periodic zero-sum bimatrix game between players $i$ and $j$ is described by a periodic sequence of payoffs where the game at time $t \geq 0$ is described by the pair of payoffs $\{A^{ij}(t), A^{ji}(t)\} = \{A(t), -A^\top(t)\}$. To obtain a counterexample, we consider the periodic zero-sum bimatrix game defined by

$$A(t) = \gamma(t)A \quad \text{where} \quad A = \begin{bmatrix} 1 & -1 \\ -1 & 1 \end{bmatrix} \quad \text{and} \quad \gamma(t) = \sin\left(\frac{2\pi t}{T}\right).$$

This game corresponds to a periodic version of matching pennies and the period of the game is $T$. The joint strategy $(x_i^*, x_j^*)$ where $x_i^* = (1/2, 1/2)$ and $x_j^* = (1/2, 1/2)$ is the unique time-invariant Nash equilibrium of the game.

Recall that in periodic zero-sum bimatrix games between players $i$ and $j$, we denote the utility of each player at time $t \geq 0$ under the joint strategy $x(t)$ by $u_i(x(t), t)$ and $u_j(x(t), t)$ to emphasize the dependence on the time-dependent payoffs which are given by $\{\gamma(t)A, -\gamma(t)A^\top\}$ in this construction. Furthermore, in this problem construction we denote the utility of each player with payoffs $\{A, -A^\top\}$ under the joint strategy $x(t)$ by $u_i(x(t))$ and $u_j(x(t))$ where $A$ is defined at the matching pennies payoff matrix defined above.

The regularization function $h_i(x_i) = \sum_{\alpha \in \mathcal{A}_i} x_{i\alpha} \log x_{i\alpha}$ in FTRL dynamics gives rise to the replicator dynamics commonly studied in evolutionary game theory. For the periodic zero-sum bimatrix game under consideration, the replicator dynamics for any strategy $\alpha \in \mathcal{A}_i$ of player $i$ are given by

$$\dot{x}_{i\alpha}(t) = x_{i\alpha}(t)\big[u_{i\alpha}(x(t), t) - u_i(x(t), t)\big] = \gamma(t)x_{i\alpha}(t)\big[u_{i\alpha}(x(t)) - u_i(x(t))\big] := \gamma(t)\dot{x}'_{i\alpha}(t).$$

Similarly, the replicator dynamics for any strategy $\alpha \in \mathcal{A}_j$ of player $j$ are given by

$$\dot{x}_{j\alpha}(t) = x_{j\alpha}(t)\big[u_{j\alpha}(x(t), t) - u_j(x(t), t)\big] = \gamma(t)x_{j\alpha}(t)\big[u_{j\alpha}(x(t)) - u_j(x(t))\big] := \gamma(t)\dot{x}'_{j\alpha}(t).$$

We now analyze the time-average of the dynamics of any strategy for each player in this game and show that they do not correspond to the time-invariant Nash equilibrium.

**Time-Average Strategies.** We begin by showing that for each player $k \in \{i, j\}$ and strategy of the player $\alpha \in \mathcal{A}_k$,

$$x_{k\alpha}\big(\tfrac{T}{2} + t\big) = x_{k\alpha}\big(\tfrac{T}{2} - t\big). \tag{33}$$

To see this, observe that for each player $k \in \{i, j\}$ and strategy of the player $\alpha \in \mathcal{A}_k$ and some initial condition $t_0$,

$$x_{k\alpha}(t) = x_{k\alpha}(t_0) + \int_{t_0}^t \dot{x}_{j\alpha}(\tau)d\tau = x_{k\alpha}(t) + \int_{t_0}^t \sin\left(\tfrac{2\pi}{T}\tau\right)\dot{x}'_{k\alpha}(\tau)d\tau. \tag{34}$$

To prove the claim in (33), we show that both $x_{k\alpha}\big(\tfrac{T}{2} + t\big)$ and $x_{k\alpha}\big(\tfrac{T}{2} - t\big)$ satisfy the same ordinary differential equation and initial condition. That is, we invoke the fundamental theorem of ordinary differential equations which says the solutions exist and are unique so that the claim holds.

Indeed, from (34) and the fundamental theorem of calculus

$$\begin{aligned}
\frac{d}{dt}x_{k\alpha}\big(\tfrac{T}{2} + t\big) &= \dot{x}'_{k\alpha}\big(\tfrac{T}{2} + t\big)\sin\left(\tfrac{2\pi}{T}\big(\tfrac{T}{2} + t\big)\right) \\
&= \dot{x}'_{k\alpha}\big(\tfrac{T}{2} + t\big)\sin\left(\pi + \tfrac{2\pi}{T}t\right) \\
&= -\dot{x}'_{k\alpha}\big(\tfrac{T}{2} + t\big)\sin\left(\tfrac{2\pi}{T}t\right).
\end{aligned}$$

Similarly,

$$\frac{d}{dt}x_{k\alpha}\left(\tfrac{T}{2}-t\right) = -\dot{x}'_{k\alpha}\left(\tfrac{T}{2}-t\right)\sin\left(\tfrac{2\pi}{T}\left(\tfrac{T}{2}-t\right)\right)$$
$$= -\dot{x}'_{k\alpha}\left(\tfrac{T}{2}-t\right)\sin\left(\pi-\tfrac{2\pi}{T}t\right)$$
$$= -\dot{x}'_{k\alpha}\left(\tfrac{T}{2}-t\right)\sin\left(\tfrac{2\pi}{T}t\right).$$

We conclude that the functions $x_{k\alpha}\left(\tfrac{T}{2}+t\right)$ and $x_{k\alpha}\left(\tfrac{T}{2}-t\right)$ satisfy the same ordinary differential equation. That is, the functional form of the ordinary differential equation is the same in both expressions. Furthermore, $x_{k\alpha}\left(\tfrac{T}{2}+0\right) = x_{k\alpha}\left(\tfrac{T}{2}-0\right) = x_{k\alpha}\left(\tfrac{T}{2}\right)$ so they satisfy the same initial condition. Hence, invoking the uniqueness property of the fundamental theorem of ordinary differential equations, the claim given in (33) holds.

The property in (33) implies for each player $k \in \{i, j\}$ and strategy of the player $\alpha \in \mathcal{A}_k$ that

$$\lim_{t\to\infty}\frac{1}{t}\int_0^t x_{k\alpha}(\tau)d\tau = \frac{1}{T}\int_0^T x_{k\alpha}(\tau)d\tau = \frac{2}{T}\int_0^{T/2} x_{k\alpha}(\tau)d\tau.$$

That is, the limiting time-average strategy is equal to the time-average strategy over half a period of the periodic game.

Now recall that the time-invariant Nash equilibrium strategy is given by the joint strategy $(x_i^*, x_j^*)$ where $x_i^* = (1/2, 1/2)$ and $x_j^* = (1/2, 1/2)$. Thus, to finish the proof we need to show for some player $k \in \{i, j\}$ and strategy of $\alpha \in \mathcal{A}_k$ that

$$\frac{2}{T}\int_0^{T/2} x_{k\alpha}(\tau)d\tau \neq \frac{1}{2}.$$

To see that the claim above holds, recall from (34) that

$$x_{k\alpha}(t) = x_{k\alpha}(0) + \int_0^t \sin\left(\tfrac{2\pi}{T}\tau\right)\dot{x}'_{k\alpha}(\tau)d\tau.$$

Observe that for any $\tau \geq 0$,

$$-1 \leq \sin\left(\frac{2\pi}{T}\tau\right) \leq 1 \quad \text{and} \quad -2 \leq \dot{x}'_{k\alpha}(\tau) \leq 2.$$

The previous expressions combine to imply

$$x_{k\alpha}(0) - 2t \leq x_{k\alpha}(t) \leq x_{k\alpha}(0) + 2t$$

and consequently

$$x_{k\alpha}(0) - \frac{T}{2} \leq \frac{2}{T}\int_0^{T/2} x_{k\alpha}(\tau)d\tau \leq x_{k\alpha}(0) + \frac{T}{2}.$$

Finally, suppose that $x_{k\alpha}(0) > 1/2$. Then, when $T < 2(x_{k\alpha}(0) - 1/2)$, the time-average of the strategy

$$\frac{2}{T}\int_0^{T/2} x_{k\alpha}(\tau)d\tau > \frac{1}{2}.$$

Analogously, suppose that $x_{k\alpha}(0) < 1/2$. Then, when $T < 2(1/2 - x_{k\alpha}(0))$, the time-average of the strategy

$$\frac{2}{T}\int_0^{T/2} x_{k\alpha}(\tau)d\tau < \frac{1}{2}.$$

Thus, unless the players initialize at the time-invariant Nash equilibrium strategy, there is a choice of the period $T$ of the periodic zero-sum matrix such that the time-average of the strategies do not converge to the time-invariant Nash equilibrium. This completes the proof.

## F Experiments

In this section, we present additional simulations and details that serve to strengthen the results in the main paper.

First, for continuous-time `GDA` dynamics we show that Poincaré recurrence holds in a periodic zero-sum bilinear game. We consider the ubiquitous Matching Pennies game with payoff matrix $A = \begin{pmatrix} 1 & -1 \\ -1 & 1 \end{pmatrix}$. We then use the following periodic rescaling with period $2\pi$:

$$\alpha(t) = \begin{cases} \sin(t) & 0 \leq t \leq \frac{3\pi}{2} \\ \left(\frac{2}{\pi}\right)(t \bmod(2\pi) - 2\pi) & \frac{3\pi}{2} \leq t \leq 2\pi \end{cases} \tag{35}$$

Hence, the bilinear zero-sum game at time $t \geq 0$ is then described by the payoffs $\{\alpha(t)A(t), -\alpha(t)A(t)^\top\}$. When agents use `GDA` learning dynamics, we see from Figure 1 that the agents' trajectories when plotted alongside the value of the periodic rescaling are bounded.

Another key result in the space of `GDA` learning dynamics is that the time-average behavior fails to converge in general to the time invariant equilibrium $(0, 0)$. A simple counterexample can be constructed by considering a periodic zero-sum bilinear game with $x_1, x_2 \in \mathbb{R}$ and a periodic rescaling $\beta(t)$ such that $\beta(t) = \beta(t + T)$ with $T = 3\pi$ for any $t \geq 0$. Moreover, let the rescaling evolve over a period as follows:

$$\beta(t) = \begin{cases} -1 & 0 \leq t \leq \pi \\ 1 & \pi \leq t \leq \frac{3\pi}{2} \\ -1 & \frac{3\pi}{2} \leq t \leq 3\pi. \end{cases} \tag{36}$$

For the simulation, we consider the payoff matrices described by $\{\beta(t)A, -\beta(t)A^\top\}$ where $A = \begin{pmatrix} 1 & -1 \\ -1 & 1 \end{pmatrix}$. We show for this example that when both players use `GDA`, the time average strategy of each player remains bounded away from the Nash $[1/2, 1/2]$, as shown in Figure 5.

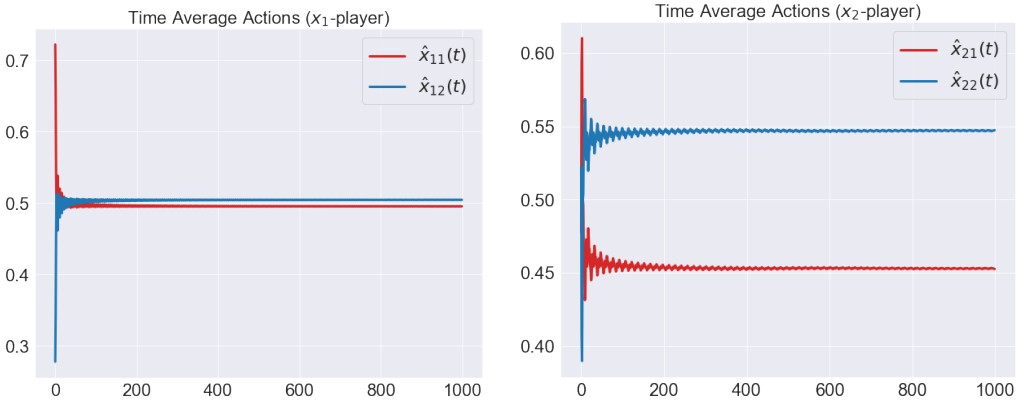

Figure 5: Time-average convergence away from the mixed Nash

**F.2** `FTRL` **Results**

For the case of `FTRL` dynamics, we perform simulations on Matching Pennies updated with replicator dynamics. The reader is reminded of the definition of replicator dynamics as a continuous analogue of multiplicative weights update, as described in Section 5. In polymatrix games, replicator dynamics for each $i \in V$ uses regularization function $h_i(x_i) = \sum_{\alpha \in \mathcal{A}_i} x_{i\alpha} \log x_{i\alpha}$ in the `FTRL` dynamics. Like the `GDA` case, we also use the periodic rescaling described in Equation 35 and obtain recurrent dynamics, as seen in Figure 6.

Theorem 3 states that the time-average utility of each player converges to the time-average value of periodic zero-sum games when each player follows `FTRL` dynamics. However, Proposition 3 states that there exist periodic zero-sum bimatrix games where the time-average strategies of `FTRL` dynamics fail to converge to the time-invariant Nash equilibrium. Here, we show a simple example that exhibits both results. Consider a Matching Pennies game that is rescaled with a $\sin$ function. Specifically, the periodic bimatrix game is given by

$$A(t) = \sin(t) \begin{bmatrix} 1 & -1 \\ -1 & 1 \end{bmatrix} \tag{37}$$

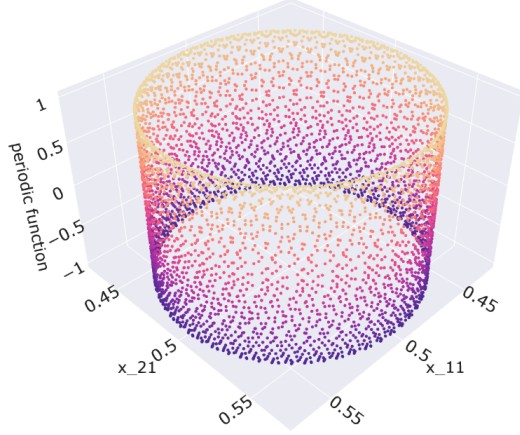

Figure 6: Periodically Rescaled Matching Pennies (Replicator)

Even with this simple example, the time-average utilities for both players go to zero, while the time-average strategies do not converge to the $[1/2, 1/2]$ time-invariant Nash, as seen in Figure 7.

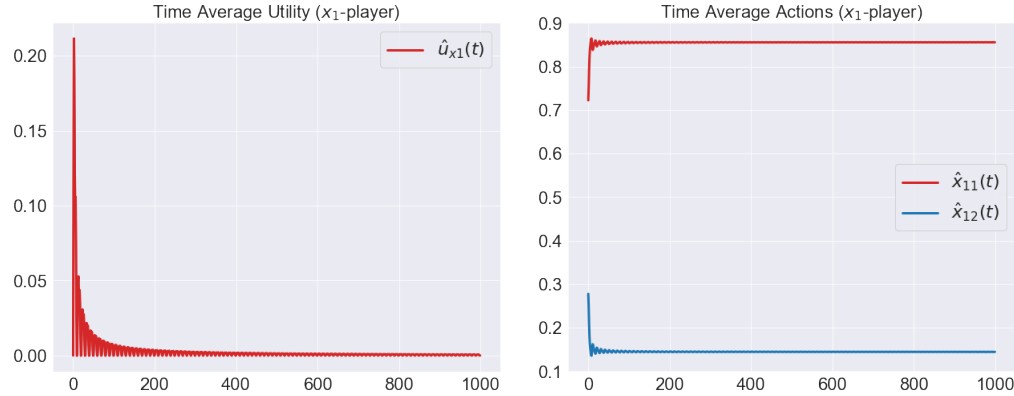

Figure 7: Time average results for MP rescaled with $sin$ function. Notice that although the average utility goes to 0, the time average actions/strategy does not go to the time invariant Nash [1/2, 1/2].

### F.3 Large Scale Simulation

We now perform simulations on larger-scale systems that present our findings in a more visually striking manner. Firstly, the time-invariant function presented in Lemma 3 and its proof can be demonstrated in any two-player periodic zero-sum polymatrix game. For replicator dynamics, the invariant function is the KL-divergence between each player's strategy and the unique mixed Nash. Using the same simulated data that was used to generate Figure 6, we show that the sum of divergences is indeed constant when both agents are using replicator dynamics. Figure 8 shows this phenomenon in the case where there are just two players playing a periodically rescaled Matching Pennies game. Specifically, the blue area represents the KL-divergence of the first player from the mixed Nash over time, and the green area represents the divergence of the second player.

To extend this formulation to the multiplayer setting, we implement a graphical polymatrix game where a number of agents are arranged in a line. Each agent then plays a bimatrix game against the agent directly adjacent to them, and the final agent also plays against the first agent. This results in a 'toroid'-like chain of games, where each agent plays against two other agents. In our simulation, each pair of agents plays the Matching Pennies game rescaled with sin against each other (Equation 37). With this system, we simulated a chain of games with 64 nodes (agents), resulting in much more complex dynamics than the two player case. Indeed, in Figure 10 we show zoomed-in plots of

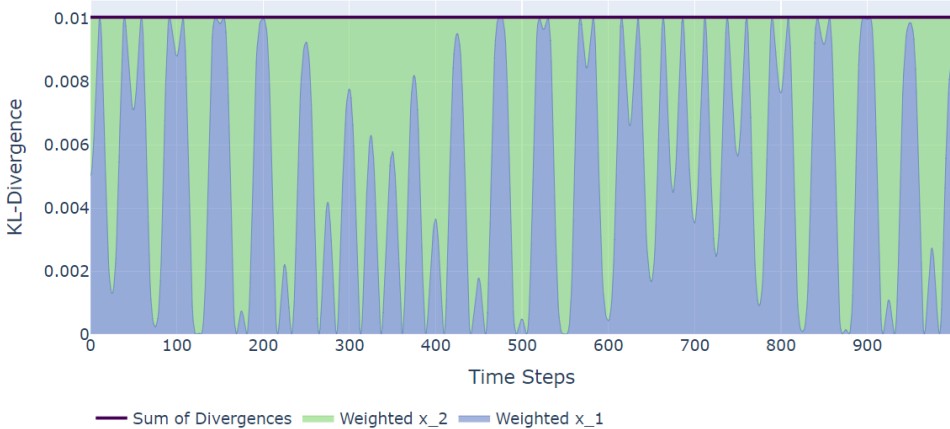

Figure 8: Time invariant function for two player periodically rescaled Matching Pennies game

Figure 2. Here, similar to previous plots, each player's KL-divergence is represented by a different-colored area. The figure showcases that on a more granular level, the individual KL-divergences of each agent can become extremely erratic, and look nowhere near periodic. Nevertheless, we see from Figure 2 that the sum of KL-divergences remains constant.

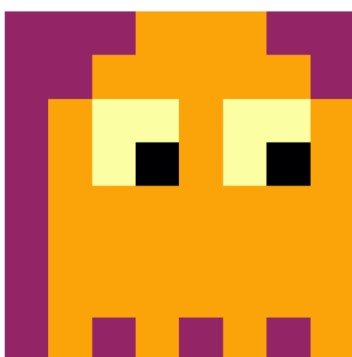

Figure 9: $8 \times 8$ grid of colors generated by sigmoid function

We also represent the trajectories of each agent by equating the strategy values of each player to RGB values in an 8x8 grid. In particular, the color of each pixel on the grid represents the probability of the respective player playing the first strategy, tuned with a sigmoid function. We then select initial conditions the correspond to RGB values such that they form the image shown in Figure 9. With the sigmoid function, any changes from the initial condition are reflected by changes in the color of the individual pixels. Thus, as agents play pairwise bimatrix games using replicator dynamics, the colors of the grid evolve. If recurrence holds, we expect to see the same image after some time. For the case of the Matching Pennies games rescaled with sin, we obtain the various images found in Figure 3, which exhibit recurrence. We also provide an animation showing how the figure evolves over time in the supplementary Jupyter notebook.

### F.4 Reproducibility Details

All experiments performed for this work were done using Python 3.7 and have been compiled into a Jupyter notebook for ease of viewing. Running the code requires only basic scientific computing packages such as NumPy and SciPy, as well as data visualization packages such as Matplotlib and Plotly. Most of the code in our submission has been edited such that it can be easily executed on a standard computer in a matter of minutes.

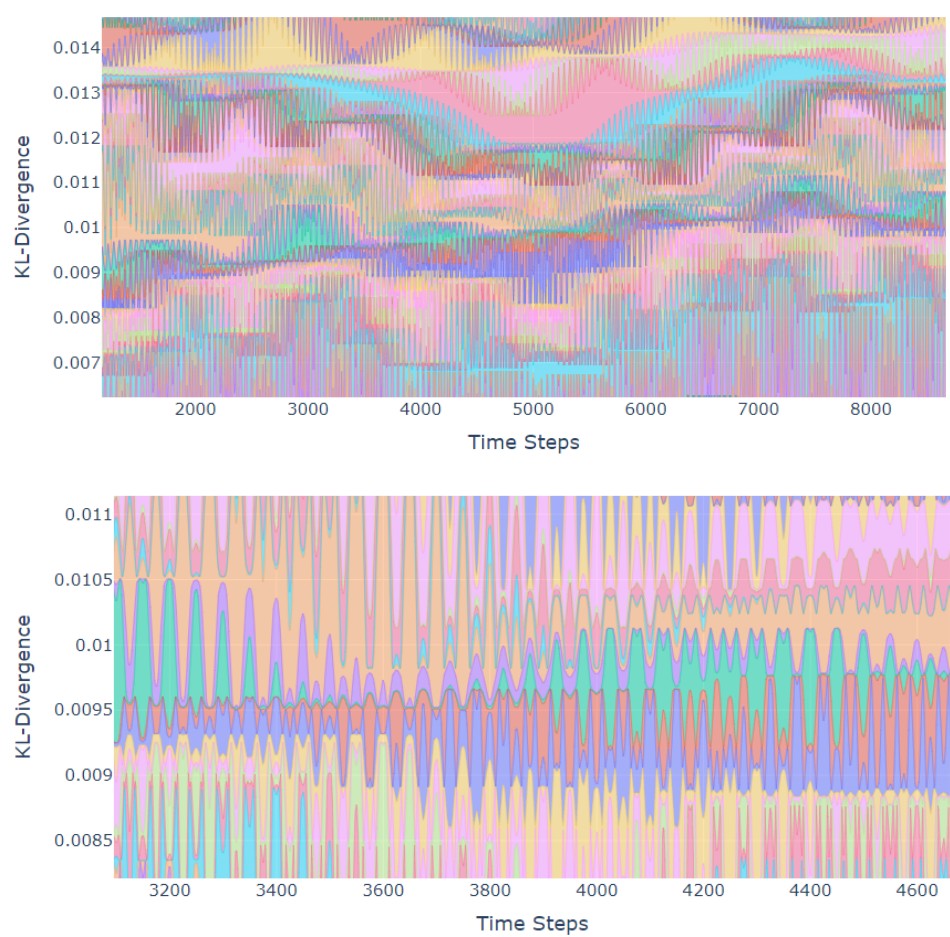

Figure 10: Zoomed-in time invariant functions for 64-player game.

## Appendix References

[1] Alexander, J.A. & Mozer, M.C. (1995) Template-based algorithms for connectionist rule extraction. In G. Tesauro, D.S. Touretzky and T.K. Leen (eds.), *Advances in Neural Information Processing Systems 7*, pp. 609–616. Cambridge, MA: MIT Press.

[2] Bower, J.M. & Beeman, D. (1995) *The Book of GENESIS: Exploring Realistic Neural Models with the GEneral NEural SImulation System.* New York: TELOS/Springer–Verlag.

[3] Hasselmo, M.E., Schnell, E. & Barkai, E. (1995) Dynamics of learning and recall at excitatory recurrent synapses and cholinergic modulation in rat hippocampal region CA3. *Journal of Neuroscience* **15**(7):5249-5262.