# OpenReview forum: "Online Learning in Periodic Zero-Sum Games"
_NeurIPS.cc/2021/Conference — NeurIPS 2021 Poster_

### Official Review · Reviewer_msZU · 2021-07-06

**Rating:** 5
**Confidence:** 4

**Summary:**

This article considers the behaviour of some modern learning in games dynamics (gradient ascent/descent and the follow the regularised leader family), applied in games that evolve periodically. The essence of the results are that if the equilibrium is constant throughout the cycle of the game, then the algorithms are periodic, whereas if the the equilibrium is not constant, or the game evolution non-periodic, then the algorithms may not be. Even in the recurrent cases, it is not always the case that the time average of strategies converge. Some simple experiments are given to demonstrate the results.

**Limitations And Societal Impact:**

Yes

**Main Review:**

The article is very nicely written, correct so far as I can tell, and interesting in a virtuous/quirky sort of way. I don't have any objections to it so far as it goes. However it is the kind of research which takes something we know (learning in fixed games) and says "how far can we push this?". The conclusion is "not very far actually" - the class of games in which the results hold is really quite restricted - zero-sum (or the network extension), a fixed equilibrium despite periodic evolution of the game, and necessarily strictly periodic game evolution. The authors try to justify this as an interesting model of the world, but I really struggle to see it. As such, I find it difficult to get excited enough about the results to commend it for presentation at NeurIPS. It's perfectly good research, just not quite exciting or relevant enough to get in.

Just one or two specific comments on the paper:

- By the start of Section 3, I'm still not sure whether time is continuous or discrete! In fact the confusion continues beyond here even. Section 1 clearly talks about discrete rounds, and Section 2 talked about having a "sequence" of games, but in Section 3.1 time is continuous, reverting back to discrete at the end of Section 3.2, then a (reasonably standard) mixture of discrete and continuous when the Poincare map is introduced. Section 4 suddenly takes the payoff functions to be defined on continuous time and we never look back. A bit more clarity/honesty would be helpful.

- Typo in Lipschitz in footnote 2.

- The proof of Thm 2 shows that the x part of the FTRL dynamics are Poincarre recurrent, following on from the z's being Poincarre recurrent. However I do not think you claim that the y are Poincarre recurrent. You could be clearer on exactly what you mean here.

- In the experimental section, it would be helpful to remind the reader that the replicator dynamics is an example of FTRL - for those of us who are not quite so familiar with this line of research, it feels a little jarring to have been talking about FTRL all article then just see replicator dynamics in the experiments.

- Caption of Fig 2. It is far from clear to me what is meant by "Weighted constant of motion". I could probably decipher it, but could the caption be clearer perhaps?


**Time Spent Reviewing:**

3

---

> ### Author Response · Authors · 2021-08-10
> **Reponse to Reviewer msZU**
>
> Thank you for your time spent reviewing and your honest feedback. We are happy to hear that you
> found the paper very nicely written!
>
> The comments you have made in the first paragraph of your review are somewhat difficult to respond to since what is exciting or relevant enough research is different person to person depending on their viewpoints and research areas. We believe that this work is certainly relevant and exciting enough to members of the community as the positive reception from the other reviewers shows. We note that as mentioned in the last paragraph of the discussion section, the broader study of evolving games has recently come into focus and shown potential, and we believe our work has a solid contribution to this area. Moreover, several of these papers have been published in AI, ML, and CS theory conferences, much like the numerous works cited in the introduction studying the emergence of Poincare recurrence in static zero-sum games. This is to say that there is an important audience at NeurIPS we believe that would appreciate this work. Thank you for your consideration of our perspective on this!
>
> Now to your specific points. We will clear up in the paper that all learning dynamics are about continuous time and make this more known up-front. Specifically, we can clear up the wording about rounds and sequences. As you pointed out, some mix between continuous and discrete is needed in Section 3, to introduce how we use the Poincare map for analysis purposes, but we will make it clear it is still for the purpose of analyzing continuous-time systems.
>
> Thanks for pointing out the typo, we will fix it.
>
> Regarding the proof of Theorem 2, we only seek to show that the x’s (strategies) of the FTRL dynamics are Poincare recurrent, since the goal is to show the dynamics are Poincare recurrent, so it is not necessary to show that the y’s (cumulative payoffs) are Poincare recurrent. We can make this more clear, namely that the y’s are an auxiliary sequence for the purpose of the FTRL update, but not part of the strategy dynamics that are being shown to be recurrent.
>
> We will be sure to remind the reader that replicator is an example of FTRL, and refer back to lines 292-295 where this was pointed out first. We can also make the caption more clear and explain the figure further. Specifically, each line in the graph represents a partial sum over the players of the KL divergence between the strategy and the invariant equilibrium, and we see that the full-sum is constant.
>
> Thanks for your consideration of our response! Please let us know if you have any remaining suggestions or if there is anything that would help you be able to recommend this paper as an accept.

---

> > ### Comment · Reviewer_msZU · 2021-08-11
> > **Response to rebuttal**
> >
> > Thank you for your detailed response. I hear you when you point out that excitement/interest is subjective. Rest assured that my discussions with the other reviewers will be open and frank and if others feel the interest is more than I do then I will not be belligerent about it.

---

### Official Review · Reviewer_mKLL · 2021-07-14

**Rating:** 8
**Confidence:** 3

**Summary:**

#  Summary of Paper
The authors study the behaviour of two continuos time
learning dynamics under two types of periodic zero-sum games, continuous gradient descent ascent (GDA) in unconstrained bilinear games,
and continuous follow the regularized leader (FTRL) in polymatrix games.
It is shown that under mild conditions, both GDA and FTRL are Point-Care recurrent,
all orbits eventually return arbitrarily close to the initial conditions.
The techniques used are common tools from dynaimcal systems but the applications to periodic games seem novel. Futhermore, strong insights on assumptions are given by providing counter examples to the Theorems, and the authors address the important issue of average convergence
in games with insightful negative and positive results.



**Limitations And Societal Impact:**

There is sufficient discussion of limitations throughout the work, however, there is no discussion on potential negative societal impact. Although no explicit section or paragraph is included in the text, the authors address the issue in their response in the checklist "The work
is primarily theoretical, and thus does not have any directly negative societal impacts." And I am in agreement with the authors, that the work is mainly theoretical without direct societal impacts.

**Main Review:**



# Originality and Quality
The theoretical results seem both novel and insightful, it is my understanding that contrary to other recent papers, this work sheds light on the behaviour of dynamics in
games that change across time whereas other works do not address time-varying games. The results are non trivial and illuminating counterexamples are provided to justify their assumptions. Additionally, the authors go beyond what is expected and address the important question of average convergence, showing that tho Pointcaré recurrence seems to transfer from fixed game to ones changing in time, a similar result does not hold for average convergence.

# Clarity
The paper is extremely well-written, making a technical topic accesible to a large audience,
though some technical details could use more clarification (as discussed below).

Although I believe the paper to be a good contribution, I have some issues regarding the technical details.
+ Where is the assumption of time-invariant Nash equilibrium used in the theorems? This seems to be hidden somewhere in the technical lemmas from the dynamical systems literature. Does it need to be unique or does the existence of one such equilibrium suffice?
+ Proposition 1 proves that both the periodic game assumption and the time-invariant equilibrium assumption are needed for the Poincare recurrent result for GDA. However I believe the second counterexample (regarding time invariant equilibrium) to be incorrect. In this example the second player is fixed, however, this implies that the dynamics no longer follow GDA, therefore the example is incompatible with the assumption of GDA dynamics.

# Suggested improvements and Clarifications

 + The assumption of a time-invariant equilibrium is mentioned in the definition of periodic polymatrix games but not in definition of periodic bilinear games, is this a mistake?
 + It is not exactly clear how volume preservation of the map $\phi^T$ and bounded orbits is used to show Pointcaré recurrence. In lines 219-221 it is argued that these conditions are sufficient for Pointcaré recurrence via Theorem 2 (on line 191), however, there is no mention of volume preservation or bounded orbits in this theorem, is this implied by the measure-preserving assumption? Or are these arguments following from line 187 (end of Theorem 1)?
+ In Poincaré recurrence theorems for maps it seems to be commonly assumed that $\phi$ is one to one (see for example V.I Arnold Section 16 D or Theorem 7.6.3 in Wiggins), why is such an assumption not needed here?
 + Why does the existence of a time invariant scalar function $\Phi$  imply bounded dynamics?
   For example the constant function would be time invariant yet this tells us nothing about the dynamics. I can see that when this is used for example in Lemma 3 that it works but generally the statement does not seem to be correct. Do you need to assume for example that $\Phi$ is coercive? For example why does the $\Phi$ in Lemma 5 work?
 + Why is zero-sum significant in polymatrix games? in general if there are more than two players then it seems that every game that is not zero-sum can be made zero-sum with the addition of a dummy player with one action and utility that is defined to be the negative sum of utilities of all other players.
 + Regarding Louiville's theorem and conservation of volume, from my understanding it is not an equivalence result, the divergence being zero implies the preservation of volume but is not a necessary condition (See for example Theorem 7.6.2 in Wiggins).

## Questions and Possible Future Work

+ There is a common reference to online learning methods, however, GDA is not a valid online method as no finite regret is guaranteed in the unconstrained setting (see for example lower bounds in Orabona). Therefore, why is GDA considered online learning?
+ Considering online mirror descent is a popular online method with continuous time analysis, it would be interesting to extensions to the case of continuous mirror descent.
+ A well-known online learning method games is that of Blackwell approachability, giving the famous regret-matching strategies (see Hart and Mas-Collel (3)). Given that there has also been continuous time analysis of Blackwelll approachability in games (Hart and Mas-Collel (4)), it would be interesting to investigate whether Pointcare recurrence appears under these dynamics as well.
+ It would be helpful to include more citations for some of the standard Dynamical systems results, for example Louiville's theorem.


# References
1. V. I. Arnol’d. Mathematical methods of classical mechanics, volume 60. Springer Science &
Business Media, 2013.
2. Wiggins, S., Wiggins, S. and Golubitsky, M., 1990. Introduction to applied nonlinear dynamical systems and chaos (Vol. 2). New York: springer-verlag.
3. Hart, S. and Mas‐Colell, A., 2000. A simple adaptive procedure leading to correlated equilibrium. Econometrica, 68(5), pp.1127-1150.
4. Hart, S. and Mas-Colell, A., 2003. Regret-based continuous-time dynamics. Games and Economic Behavior, 45(2), pp.375-394.
5. Orabona, F., 2019. A modern introduction to online learning. arXiv preprint arXiv:1912.13213.

**Time Spent Reviewing:**

10

---

> ### Author Response · Authors · 2021-08-10
> **Response to Reviewer mKLL**
>
> Thank you for your extremely detailed review and the amount of time you put into giving us feedback on our paper, it is greatly appreciated! We also would like to thank you for your positive comments, and the general tone of your review being aimed at improving our work.
>
> **Clarity:**
>
> The primary way the time-invariant equilibrium is used is in proving that orbits are bounded. In particular, the time-invariant functions essentially are constructed by showing that some measure of energy between the dynamics or a transformation of the dynamics and the equilibrium remain constant, which relies on the equilibrium being time-invariant.  Note that this is used in constructing the time-invariant function for the GDA dynamics since (0,0) is a time-invariant equilibrium. The time-invariant functions are then used to conclude that the orbits must be bounded. The time-invariant equilibrium does not need to be unique, it just needs to exist. This is similar to past results on static zero-sum polymatrix games that simply require the existence of an interior equilibrium for recurrence (see e.g., Cycles in Adversarial Learning, Mertikopoulos et al. 2018). We can make this more clear in the paper, thank you.
>
> Thank you for your point about Proposition 1 and Example 2. It is true that in some sense it is not GDA, because it is essentially just a degenerate game in that there is only a single-player. The reason we presented the result this way was that it seemed it would be more clean to present both parts of Proposition 1 using the same setting, namely GDA in the unconstrained time-evolving bilinear zero-sum games. However, since in this setting (0,0) will always be a time-invariant equilibrium it is difficult to do without going to the setting we have presented. If the example we have provided is not satisfying to you, we can further add the following example for case of bimatrix (two-player polymatrix) zero-sum setting with an instantiation of the FTRL dynamics.  In particular, consider a setting where the payoff matrix for the first 1/4 of the period is matching pennies with $A= [1, -1; -1, 1]$ and for the last 3/4 of the period it is $A= [0.05, -0.5; -0.5, 5]$. Note that here the payoff matrix for the second player is just -A. The former game has a mixed Nash equilibrium of $x_1^{\ast}=x_2^{\ast}=(0.5, 0.5)$ and the latter game has a mixed Nash equilibrium of $x_1^{\ast}=x_2^{\ast}=(0.9091,0.0909)$. We simulate this example with replicator dynamics, which is an instantiation of the FTRL dynamics. We plot the trajectories of the player’s coordinates against each other. Moreover, we plot the $\ell_1$ norm distance between the joint trajectory of the players and the initial condition. The simulation results show that the trajectory does not return back arbitrarily close to the initial condition, so the dynamics are not Poincare recurrent in this periodic evolving game without a time-invariant equilibrium. We are happy to add an example of this nature if you feel that it helps. Note that as specified by the NeurIPS guidelines at https://nips.cc/Conferences/2021/PaperInformation/NeurIPS-FAQ, if necessary anonymous links can be provided to answer questions. If you feel you need to view the actual simulation results, the plots can be viewed at the following double-blind anonymous link  https://anonymous.4open.science/r/rebuttal_figs-E482/.
>
>
> **Suggested Improvements and Clarifications.**
>
> Thank you for these points, they are all extremely helpful for improving the clarity of our paper.
>
> We did not include that assumption since (0,0) will be a time-invariant equilibrium in the two-player zero-sum bilinear games, but we will make that more clear.
>
> Regarding your 2nd point, yes the volume preservation property and the bounded orbits implies the measure preserving property. We will make this more concrete.
>
> The assumption that the map is one to one is not needed here because the conditions of the learning dynamics are satisfied such that the fundamental theorem of differential equations ensures that the solution exists and is unique so that the map is one to one. This is somewhat pointed out in footnote 2, but we can make it more clear in the paper.
>
> As you pointed out, the existence of a time-invariant function on its own cannot imply bounded orbits immediately. The terminology used in the paper was intended to note that bounded orbits is proven by constructing a time-invariant function that implies this property. As you noted, this is more clear in the GDA case. The fact that the time-invariant function implies bounded orbits in the FTRL case is time-invariant is more complicated, but we noted at the end of the proof of Lemma 5 that this follows identically from Lemma D.2 in (Cycles in Adversarial Learning, Mertikopoulos et al. 2018). If you would like we can add this argument for the sake of completeness.
>
> We do not think the construction you have argued about making polymatrix games zero-sum is generally possible because of how polymatrix games are structured. Namely, the utility for a player is the sum of the utilities over each bimatrix game that the player is in, and a fixed strategy is used across all the games. So it is not generally possible to add a player in that way with a dummy action achieving that goal (without defining the payoff matrix to be a function of other players actions). Let us know if we are missing something in what you suggested.
>
> Regarding Liouville's Theorem, in some cases it does go both directions, but nonetheless we only need the direction you have pointed out, so we will clarify this and be more precise.
>
>
> **Questions and Possible Future Work:**
>
> Thank you for these great suggestions!
>
> The definition of online learning could be argued is not universally defined; the terminology in this paper mainly comes from these being classical sets of online update methods for learning in games. The reference you have provided is also a bit different since it appears those results are simply for unconstrained linear optimization against an adversary (assuming this is what you are referring to in the reference you have provided), so there is not the additional structure from each player in the game doing gradient ascent on their utility function. See [1] for more on regret of GDA and variants in unconstrained static zero-sum bilinear games.
>
> [1] Gidel et al. Finite Regret and Cycles with Fixed Step-Size via Alternating Gradient Descent-Ascent. COLT 2020.
>
> Regarding mirror descent and FTRL, often times these are equivalent (see e.g., chapter 2 of Online Learning and Online Convex Optimization, Shai Shalev-Shwartz). The connection with the regret-matching strategies and Blackwell approachability is quite interesting, and worth looking into since we are not aware of any results in the context of recurrence for them.
>
> Thank you for the pointer to add more citations to dynamical systems results, we can certainly do that.

---

### Official Review · Reviewer_a6SU · 2021-07-16

**Rating:** 7
**Confidence:** 3

**Summary:**

The paper aims to analyze online learning behaviors of the gradient descent-ascent(GDA) and the follow the regularized leader(FTRL) algorithms for periodic zero-sum games.
Seminal works [23, 19, 7] show that online no-regret learning dynamics is Poincar\'e recurrent in repeated static zero-sum games, but the techniques used in the seminal works does not work well in periodic one. To resolve this, the paper exploits an important theorem about volume preservation for periodic systems by [1], and shows that GDA dynamics in periodic zero-sum bilinear games and FTRL in periodic strategy polymatrix zero-sum games are Poincar\'e recurrent. The paper also shows a negative result that time-average strategy does not converge to time-invariant equilibrium, as a counter example. The empirical evaluation support the theoretical results.
(Reference number follows the paper.)

**Limitations And Societal Impact:**

As described in the significance, periodic zero-sum games are limited to dealing with the evolution over time and have few applications. However, I think the contributions of the paper outweigh this limit.


**Main Review:**

Originality:
Whereas the no-regret learning algorithm dynamics is known to be Poincar\'e recurrent in static zero-sum games , to the best of my knowledge, this is the first research showing that GDA and FTRL learning dynamics are Poincare recurrence in periodic zero-sum games.

Quality:
Proposed theorems and lemmas seem to be technically sound, and the experimental results support the theorems.

Clarity:
The paper is well-written and easy to follow, and honestly describe a negative result that there exists a counter example.

Significance:
It is significant to prove that GDA and FTRL learning dynamics are Poincar\'e recurrent in some games. Whereas periodic zero-sum games seem to have few applications, the proposed theorems are an important first step to explore dynamics in other games evolving over time.


**Time Spent Reviewing:**

4hours

---

> ### Author Response · Authors · 2021-08-10
> **Response to Reviewer a6SU**
>
> Thank you for your time spent reviewing and the valuable feedback you have provided!

---

### Official Review · Reviewer_yvC8 · 2021-07-28

**Rating:** 7
**Confidence:** 3

**Summary:**

In this paper, the authors aim to analyze the asymptotic behavior of online learning algorithms in periodic rather than static zero-sum games. In terms of positive results, they show that the time-evolving *periodic* versions of both zero-sum bilinear games (under Gradient Descent Ascent) and zero-sum polymatrix games with time-invariant equilibria (under FTRL) satisfy the notion of Poincare recurrence -- a formal guarantee of cyclic behavior. However, the authors provide negative results too: in the described cases, the time-average of the GDA/FTRL plays does not converge to a time-invariant Nash equilibrium (in the latter case there is utility convergence).

**Main Review:**

This work has sufficient clarity of exposition, technical contributions, and insights for a publication. However, I do have high level questions/concerns about the work itself. The particular choice of GDA and FTRL for the two game variants was not substantiated -- why not GDA in both cases for example? Since the results (negative and positive) are particular to these algorithms, it would be good to know how general the treatment is (would such arguments work for other algorithms). More importantly, given that the goal is to investigate time-evolving games, is there a reason to believe that Poincare recurrence or time-invariant equilibria are the right measures of performance. Finally, what can be said in terms of finite time convergence, computational efficiency, etc? (for future work)

At the same time, this work shows that there are many unanswered questions in this direction (some of which the authors answer here). I would like to see the discussion section to actually be about discussion -- what was answered and what else should be investigated. And please add a related works section (some is in section 7) after the intro (or subsection in intro). I am rating this submission as a weak accept which I'll upgrade to accept if there are no major issues found by other reviewers.

Minor comments: in Definition 1, there is no assumption about a time-invariant equilibrium, I'm guessing because (0, 0) is always one. This is a very trivial case in my opinion, and I think it's worth it to consider constrained action sets so that there is no obvious equilibrium one knows (Poincare recurrence is still nontrivial but time-average convergence of, say, GDA does not say as much).

The negative results in sections 4, 5 are specific to GDA/FTRL so they don't imply "that the time-evolving game model of periodic zero-sum games is at the cusp of complexity where positive results are achievable". This just makes the results in the paper regarding GDA, FTRL stronger as the set of assumptions for them is more or less tight.

**Time Spent Reviewing:**

8

---

> ### Author Response · Authors · 2021-08-10
> **Response to Reviewer yvC8**
>
> Thank you for your time spent reviewing and the valuable feedback you have provided.
>
> The choices of studying GDA and FTRL in the respective classes of games come about from each being common, `natural’ sets of learning dynamics and basic staples of online optimization theory. Moreover, we remark that FTRL covers a broad set of learning dynamics dependent on the choice of regularization function used in the dynamics, so it is best viewed as a family of algorithms (see lines 292-295 for some common regularizers). So, at least in the case of time-evolving zero-sum polymatrix games, the treatment is fairly general. In fact, as is mentioned in lines 292-295, the projected gradient dynamics are an instantiation of the FTRL dynamics, so we are effectively studying the gradient dynamics in each class of games. We hope this resolves your comments in the start of your review about the algorithms and we will happily add more commentary on why these sets of dynamics are studied.
>
> It is an interesting philosophical point you bring up about Poincare recurrence and measures of performance. It is not so much that seeking Poincare recurrence is the goal (although it is a common dynamical systems notion of regularity in the behavior of the dynamics), but rather it simply emerges from these sets of learning dynamics in the classes of games that are studied. To your question about convergence, one interesting direction of future work would be looking to see if so-called optimistic versions of FTRL, which in certain classes of static zero-sum games have last iterate convergence guarantees to Nash equilibrium, also exhibit this behavior in time-evolving games with invariant equilibrium. Thank you for your suggestion regarding the discussion section, and moving more related work earlier in the paper. We will certainly make these revisions. We appreciate the points that you have brought up, which will be great to incorporate into the discussion.
>
> Regarding your ``minor comments’’: yes that is because (0,0) is always one. We can make this more explicit in the definition. See the above parts of the response which address your question about constrained action sets; Specifically, players are constrained to the simplex in polymatrix games and the FTRL dynamics always lie on the simplex.
>
> We will remove the sentence ``..at the cusp..’’ and adjust accordingly based on your suggestion.
>
> Thanks again for your work!

---

### Author Response · Authors · 2021-08-26
**Thanks to the reviewers: we are available if any questions remain**

Thank you to the reviewers for their hard work. We just wanted to check in and see if there was any feedback on our responses and also if there is any lingering questions that we could help answer.

---

### Decision · Program_Chairs · 2021-09-27

**Decision:**

Accept (Poster)

**Comment:**

No Reviewer has major concerns on this paper. The only objection is raised by Reviewer msZU and concerns the motivation behind the model and the significance. Such Reviewer also raises other points. I invite the authors to address these points carefully, together with the motivation behind the model, in the final version of the paper.